# Enhancing Vector Quantization with Distributional Matching: A Theoretical and Empirical Study

## Abstract

The success of autoregressive models largely depends on the effectiveness of vector quantization, a technique that discretizes continuous features by mapping them to the nearest code vectors within a learnable codebook. Two critical issues in existing vector quantization methods are training instability and codebook collapse. Training instability arises from the gradient discrepancy introduced by the straight-through estimator, especially in the presence of significant quantization errors, while codebook collapse occurs when only a small subset of code vectors are utilized during training. A closer examination of these issues reveals that they are primarily driven by a mismatch between the distributions of the features and code vectors, leading to unrepresentative code vectors and significant data information loss during compression. To address this, we employ the Wasserstein distance to align these two distributions, achieving near 100% codebook utilization and significantly reducing the quantization error. Both empirical and theoretical analyses validate the effectiveness of the proposed approach.

## 1 Introduction

Autoregressive models have re-emerged as a powerful paradigm in visual generation, demonstrating significant advances in image synthesis quality. Recent studies [29, 9, 6, 19, 35, 20] highlight that autoregressive approaches now achieve superior results compared to diffusion-based methods [12, 30, 33, 35, 24]. The success of autoregressive visual generative models hinges on the effectiveness of vector quantization (VQ) [36], a technique that compresses and discretizes continuous features by mapping them to the nearest code vectors within a learnable codebook. However, VQ continues to face two major challenges: training instability and codebook collapse.

The first issue originates from the non-differentiability of VQ, which prevents direct gradient backpropagation from quantized features to their continuous counterparts, thereby hindering effective model optimization. To address this challenge, VQ-VAE [36] introduces a straight-through estimator (STE) [2]. The STE facilitates gradient propagation by copying the gradients from the quantized features to the continuous features. Nevertheless, the effectiveness of this approach is critically contingent upon the magnitude of the quantization error between the continuous and quantized feature vectors. When the quantization error is excessively large, the training process becomes notably unstable [19].

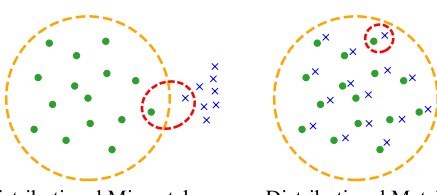

Distributional Mismatch      Distributional Match

Figure 1: The symbols $\cdot$ and $\times$ represent the feature and code vectors, respectively. The left figure illustrates the distributional mismatch between the feature and code vectors, while the right figure visualizes their distributional match.

Submitted to 39th Conference on Neural Information Processing Systems (NeurIPS 2025). Do not distribute.

The latter issue emerges due to the inability of existing VQ methods to ensure that all Voronoi cells[1] are assigned feature vectors. When only a minority of Voronoi cells are allocated feature vectors, leaving the majority unutilized and unoptimized, severe codebook collapse ensues [42]. Despite considerable research efforts dedicated to mitigating this problem, these methods still exhibit relatively low utilization of code vectors, particularly in scenarios with large codebook sizes [8, 34, 39, 19, 42]. This is due to the fact that, as the codebook size increases, the number of Voronoi cells also increases, significantly raising the challenge of ensuring that every cell is assigned a feature vector.

In this paper, we examine these issues by investigating the distributions of the features and code vectors. To illustrate the idea, Figure 1 presents two extreme scenarios: the left panel depicts a significant mismatch between the two distributions, while the right panel shows a match. In the left panel, all features are mapped to a single codeword, resulting in large quantization errors and minimal codebook utilization. In contrast, the right panel demonstrates that a distributional match leads to negligible quantization error and near 100% codebook utilization. This suggests aligning these two distributions in VQ could potentially address the issues of training instability and codebook collapse.

To investigate the idea above, we first introduce three principled criteria that a VQ method should satisfy. Guided by this criterion triple, we conduct qualitative and quantitative analyses, demonstrating that aligning the distributions of the feature and code vectors results in near 100% codebook utilization and minimal quantization error. Additionally, our theoretical analysis underscores the importance of distribution matching for vector quantization. To achieve this alignment, we employ the quadratic Wasserstein distance which has a closed-form representation under a Gaussian hypothesis. Our approach effectively mitigates both training instability and codebook collapse, thereby enhancing image reconstruction performance in visual generative tasks.

## 2 Understanding Distribution Matching

This section introduces a novel distributional perspective for VQ. By defining three principled criteria for VQ evaluation, we empirically and theoretically demonstrate that distribution matching yields an almost optimal VQ solution.

### 2.1 An Overview of Vector Quantization

As the core component in visual tokenizer [36, 19, 35], VQ acts as a compressor that discretizes continuous latent features into discrete visual tokens by mapping them to the nearest code vectors within a learnable codebook.

Figure 2 illustrates the classic VQ process [36], which consists of an encoder $E(\cdot)$, a decoder $D(\cdot)$, and an updatable codebook $\{\mathbf{e}_k\}_{k=1}^{K} \in \mathbb{R}^d$ containing a finite set of code vectors. Here, $K$ represents the size of the codebook, and $d$ denotes the dimension of the code vectors. Given an image $\boldsymbol{x} \in \mathbb{R}^{H \times W \times 3}$, the goal is to derive a spatial collection of codeword IDs $r \in \mathbb{N}^{h \times w}$ as image tokens. This is achieved by passing the image through the encoder to obtain

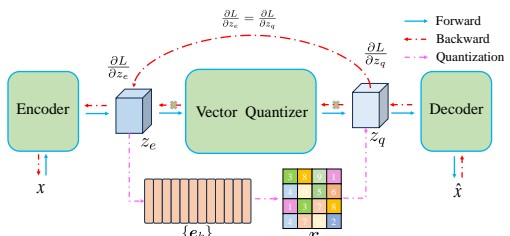

Figure 2: The illustration of VQ.

$\boldsymbol{z}_e = E(\boldsymbol{x}) \in \mathbb{R}^{h \times w \times d}$, followed by a spatial-wise quantizer $\mathcal{Q}(\cdot)$ that maps each spatial feature $\boldsymbol{z}_e^{ij}$ to its nearest code vector $\boldsymbol{e}_k$:

$$r^{ij} = \arg\min_{k} \|\boldsymbol{z}_e^{ij} - \boldsymbol{e}_k\|_2^2. \tag{1}$$

These tokens are then used to retrieve the corresponding codebook entries $\boldsymbol{z}_q^{ij} = \mathcal{Q}(\boldsymbol{z}_e^{ij}) = \boldsymbol{e}_{r^{ij}}$, which are subsequently passed through the decoder to reconstruct the image as $\widehat{\boldsymbol{x}} = D(\boldsymbol{z}_q)$. Despite its success in high-fidelity image synthesis [36, 29, 9], VQ faces two key challenges: training instability and codebook collapse.

---

[1]A comprehensive understanding of codebook collapse through the lens of Voronoi partition is provided in Appendix C.

**Training Instability** This issue occurs because during backpropagation, the gradient of $z_q$ cannot flow directly to $z_e$ due to the non-differentiable function $\mathcal{Q}$. To optimize the encoder's network parameters through backpropagation, VQ-VAE [36] employs the straight-through estimator (STE) [3], which copies gradients directly from $z_q$ to $z_e$. However, this approach carries significant risks—especially when $z_q$ and $z_e$ are far apart. In these cases, the gradient gap between the representations can grow substantially, destabilizing the training process. In this paper, we tackle the training instability challenge from a distributional viewpoint.

**Codebook Collapse** Codebook collapse occurs when only a small subset of code vectors receives optimization-useful gradients, while most remain unrepresentative and unupdated [8, 34, 39, 19, 42]. Researchers have proposed various solutions to this problem, such as improved codebook initialization [43], reinitialization strategies [8, 38], and classical clustering algorithms like $k$-means [5] and $k$-means++[1] for codebook optimization [29, 42]. Beyond these deterministic approaches that select the best-matching token, researchers have also explored stochastic quantization strategies [40, 28, 34].

However, these methods still exhibit relatively low utilization of code vectors, particularly with large codebook sizes $K$ [42, 25]. In this paper, we address this issue by the distributional matching between feature vectors and code vectors.

## 2.2 Evaluation Criteria

Given a set of feature vectors $\{z_i\}_{i=1}^N$ sampled from feature distribution $\mathcal{P}_A$ and code vectors $\{e_k\}_{k=1}^K$ sampled from codebook distribution $\mathcal{P}_B$, vector quantization involves finding the nearest, and thus most representative, code vector for each feature vector:

$$z_i' = \underset{e \in \{e_k\}}{\arg\min} \|z_i - e\|.$$

The original feature vector $z_i$ is then quantized to $z_i'$. Below, we introduce three key criteria to evaluate this process.

**Criterion 1** (Quantization Error). *The quantization error measures the average distortion introduced by VQ and is defined as*

$$\mathcal{E}(\{e_k\}; \{z_i\}) = \frac{1}{N} \sum_i \|z_i - z_i'\|^2.$$

A smaller $\mathcal{E}$ signifies a more accurate quantization of the original feature vectors, resulting in a smaller gradient gap between $z_i$ and $z_i'$. Consequently, a small $\mathcal{E}$ suggests that the issue of training instability can be effectively mitigated.

**Criterion 2** (Codebook Utilization Rate). *The codebook utilization rate measures the proportion of code vectors used in VQ and is defined as*

$$\mathcal{U}(\{e_k\}; \{z_i\}) = \frac{1}{N} \sum_{i=1}^N \mathbf{1}(e_k = z_i' \text{ for some } i).$$

A higher value of $\mathcal{U}$ reduces the risk of codebook collapse. Ideally, $\mathcal{U}$ should reach 100%, indicating that all code vectors are utilized. As discussed in Appendix D, $\mathcal{U}$ can only measure the *completeness* of codebook utilization; it does not suffice to evaluate the degree of codebook collapse. This motivates us to introduce the codebook perplexity criterion.

**Criterion 3** (Codebook Perplexity). *The codebook perplexity measures the uniformity of codebook utilization in VQ and is defined as*

$$\mathcal{C}(\{e_k\}; \{z_i\}) = \exp(-\sum_{k=1}^K p_k \log p_k),$$

where $p_k = \frac{1}{N} \sum_{i=1}^N \mathbf{1}(z_i' = e_k)$. A higher value of $\mathcal{C}$ indicates that code vectors are more uniformly selected in the VQ process. Ideally, $\mathcal{C}$ reaches its maximum at $\mathcal{C}_0 = \exp(-\sum_{k=1}^K \frac{1}{K} \log \frac{1}{K}) = K$ when code vectors are completely uniformly utilized. Therefore, as a complementary measure to Criterion 2, the combination of $\mathcal{U}$ and $\mathcal{C}$ can effectively evaluate the degree of codebook collapse.

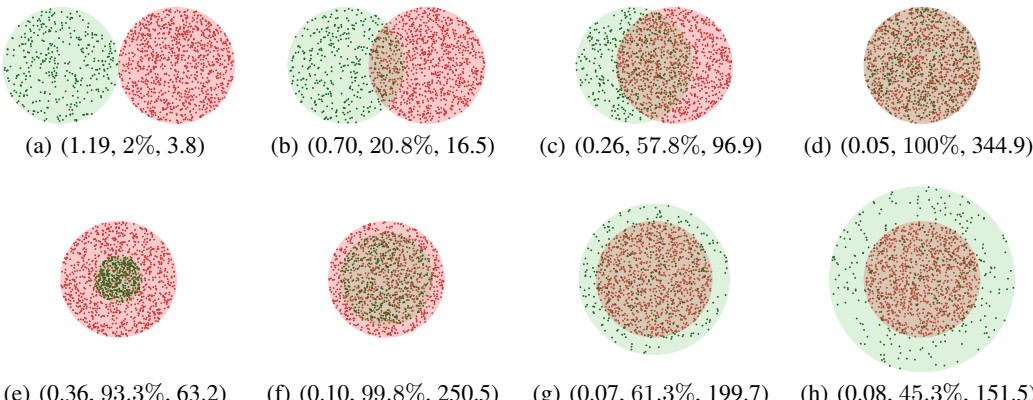

(a) (1.19, 2%, 3.8)   (b) (0.70, 20.8%, 16.5)   (c) (0.26, 57.8%, 96.9)   (d) (0.05, 100%, 344.9)

(e) (0.36, 93.3%, 63.2)   (f) (0.10, 99.8%, 250.5)   (g) (0.07, 61.3%, 199.7)   (h) (0.08, 45.3%, 151.5)

Figure 3: Qualitative analyses of the criterion triple $(\mathcal{E}, \mathcal{U}, \mathcal{C})$: The red and green disks represent the uniform distributions of feature vectors and code vectors, respectively.

We refer to $(\mathcal{E}, \mathcal{U}, \mathcal{C})$ as the criterion triple. When comparing extreme cases of distributional match and mismatch shown in Figure 1, we find that distributional matching significantly outperforms mismatching across all three criteria. Using this criterion triple, we present detailed analyses that demonstrate the advantages of distribution matching.

## 2.3 The Effects of Distribution Matching

We conduct a simple synthetic experiment to provide intuitive insights (See experimental details in Appendix I.1). Specifically, we assume that the distributions $\mathcal{P}_A$ and $\mathcal{P}_B$ are uniform distributions confined within two distinct disks, as depicted in Figure 3. We then sample a set of feature vectors $\{z_i\}_{i=1}^N$ uniformly from the red disk, and a set of code vectors $\{e_k\}_{k=1}^K$ uniformly from the green circle. The criterion triple $(\mathcal{E}, \mathcal{U}, \mathcal{C})$ is then calculated based on the definitions in Criteria 1 to 3.

We examine two cases. The first involves two disks with identical radii but different centers. As shown in Figures 3(a) to 3(d), when the centers of the disks move closer together, the criterion triple improves toward optimal values. Specifically, $\mathcal{E}$ decreases from 1.19 to 0.05, $\mathcal{U}$ rises from 2% to 100%, and $\mathcal{C}$ increases from 3.8 to 344.9.

The second case shows two distributions with identical centers but different radii. When the codebook distribution's support lies within the feature distribution's support (as shown in Figures 3(e) and 3(f)), it results in a notably larger $\mathcal{E}$, slightly lower $\mathcal{U}$, and significantly smaller $\mathcal{C}$ compared to the aligned distributions shown in Figure 3(d). Conversely, when the codebook distribution's support extends beyond the feature distribution's support, $\mathcal{E}$ shows a modest increase while both $\mathcal{U}$ and $\mathcal{C}$ decrease significantly, as illustrated in Figures 3(g) and 3(h). We provide detailed explanations of these experimental results in Appendix E.

From both cases, we can conclude that the VQ achieves the optimal criterion triple when the feature and codebook distributions are identical. This observation will be further supported by more quantitative analyses in Appendix F.

## 2.4 Theoretical Analyses

In this section, we provide theoretical evidence to support our empirical observations. Let the code vectors $\{e_k\}_{k=1}^K$ and feature vectors $\{z_i\}_{i=1}^N$ be independently and identically drawn from $\mathcal{P}_B$ and $\mathcal{P}_A$, respectively. We say a codebook $\{e_k\}_{k=1}^K$ attains full utilization asymptotically with respect to $\{z_i\}_{i=1}^N$ if the codebook utilization rate $\mathcal{U}(\{e_k\}_{k=1}^K; \{z_i\}_{i=1}^N)$ tends to 1 in probability as $N$ approaches infinity:

$$\mathcal{U}(\{e_k\}_{k=1}^K; \{z_k\}_{i=1}^N) \xrightarrow{p} 1, \quad \text{as } N \to \infty.$$

For the codebook distribution $\mathcal{P}_B$, we say it attains full utilization asymptotically with respect to $\mathcal{P}_A$ if, with probability 1, the randomly generated codebook $\{e_k\}_{k=1}^K$ achieves full utilization asymptotically.

Additionally, a codebook distribution $\mathcal{P}_B$ is said to have vanishing quantization error asymptotically with respect to a domain $\Omega \subseteq \mathbb{R}^d$ if the quantization error over all data of size $N$ tends to zero in probability as $K$ approaches infinity:

$$\sup_{\{z_i\} \subseteq \Omega} \mathcal{E}(\{e_k\}_{k=1}^K; \{z_i\}_{i=1}^N) \xrightarrow{p} 0, \quad \text{as } K \to \infty. \tag{2}$$

Our first theorem shows that $\overline{\mathrm{supp}(\mathcal{P}_A)} = \overline{\mathrm{supp}(\mathcal{P}_B)}$ is sufficient and necessary for the codebook distribution $\mathcal{P}_B$ to attain both full utilization and vanishing quantization error asymptotically. For simplicity, $\mathcal{P}_A$ is assumed to have a density function $f_A$ with bounded support $\Omega \subseteq \mathbb{R}^d$.

**Theorem 1.** *Assume $\Omega = \mathrm{supp}(\mathcal{P}_A)$ is a bounded open area. The codebook distribution $\mathcal{P}_B$ attains full utilization and vanishing quantization error asymptotically if and only if $\overline{\mathrm{supp}(\mathcal{P}_B)} = \overline{\mathrm{supp}(\mathcal{P}_A)}$, where $\overline{\mathcal{S}}$ denotes the closure of the set $\mathcal{S}$.*

Theorem 1 establishes the optimal support of the codebook distribution. The boundedness of $\Omega$ is required as we consider the worst case quantization error in equation 2. In real applications, when $\mathcal{P}_A$ follows an absolutely continuous distribution over an unbounded domain, then $\{z_i\}_{i=1}^N$ generated from $\mathcal{P}_A$ will be bounded with high probability. Thus, Theorem 1 also provides theoretical insights for a target distribution $\mathcal{P}_A$ with an unbounded domain.

Besides the optimal support, we also determine the optimal density of the codebook distribution by invoking existing results characterizing asymptotic optimal quantizers [10]. Specifically, we consider the case where $N$ approaches to infinity and define the expected quantization error of a codebook $\{e_k\}$ with respect to $\mathcal{P}_A$ as

$$\mathcal{E}(\{e_k\}_{k=1}^K; \mathcal{P}_A) = \mathbb{E}_{z \sim \mathcal{P}_A} \min_{e \in \{e_k\}} \|z - e\|^2.$$

A codebook $\{e_k^*\}_{k=1}^K$ is called the set of optimal centers for $\mathcal{P}_A$ if it achieves the minimal quantization error:

$$\mathcal{E}(\{e_k^*\}_{k=1}^K; \mathcal{P}_A) = \min_{\{e_k\}_{k=1}^K} \mathcal{E}(\{e_k\}_{k=1}^K; \mathcal{P}_A).$$

Theorem 2 demonstrates that, under weak regularity conditions, the empirical measure of the optimal centers for $\mathcal{P}_A$ converges in distribution to a fixed distribution determined by $\mathcal{P}_A$. Notably, we do not assume a bounded domain in the following theorem.

**Theorem 2** (Theorem 7.5, [10]). *Suppose $Z \sim \mathcal{P}_A$ is absolutely continuous with respect to the Lesbegue measure in $\mathbb{R}^d$ and $\mathbb{E}\|Z\|^{2+\delta} < \infty$ for some $\delta > 0$. Then the empirical measure of the optimal centers for $\mathcal{P}_A$,*

$$\frac{1}{K} \sum_{k=1}^K \delta_{e_k^*},$$

*converges weakly to a fixed distribution $\mathcal{P}_A^*$, whose density function $f_A^*$ is proportional to $f_A^{(d+2)/d}$.*

Theorem 2 implies that $\mathcal{P}_B = \mathcal{P}_A^*$ is the optimal codebook distribution in the asymptotic regime as $K$ approaches infinity. In high-dimensional spaces with large $d$, this optimal distribution $\mathcal{P}_B = \mathcal{P}_A^*$ closely approximates $\mathcal{P}_A$. This further motivates us to align the codebook distribution $\mathcal{P}_B$ with the feature distribution $\mathcal{P}_A$.

## 3 Methodology

In this section, we introduce the quadratic Wasserstein distance for distributional matching between features and the codebook. We then apply this technique to two frameworks.

### 3.1 Distribution Matching via Wasserstein Distance

We assume a Gaussian hypothesis for the distributions of both the feature and code vectors. For computational efficiency, we employ the quadratic Wasserstein distance, as defined in Appendix B, to align these two distributions. Although other statistical distances, such as the Kullback-Leibler divergence [17, 12], are viable alternatives, they lack simple closed-form representations, making them computationally expensive. The following lemma provides the closed-form representation for the quadratic Wasserstein distance between two Gaussian distributions.

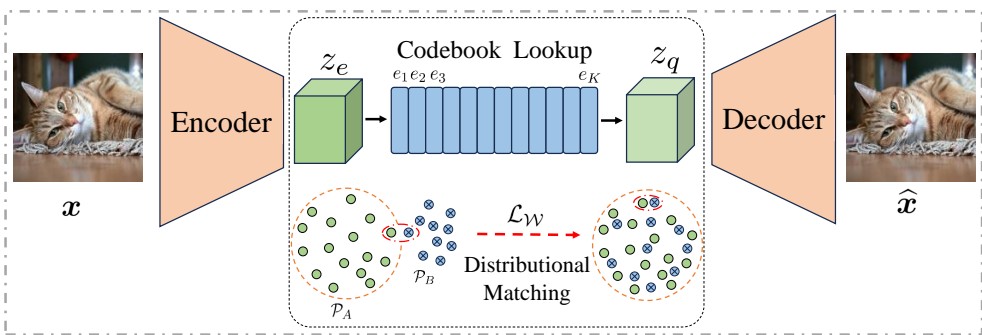

Figure 4: Illustration of the *Wasserstein VQ*. The architecture integrates an encoder-decoder network with a VQ module. In the VQ module, we augment the vanilla VQ framework [36] by incorporating our proposed Wasserstein loss $\mathcal{L}_{\mathcal{W}}$ to achieve distributional matching between features $\boldsymbol{z}_e$ ($\boldsymbol{z}_e^{ij} \sim \mathcal{P}_A$) and the codebook $\boldsymbol{e}_k$ ($\boldsymbol{e}_k \sim \mathcal{P}_B$). This enhancement leads to 100% codebook utilization and the minimal achievable quantization error between $\boldsymbol{z}_e$ and $\boldsymbol{z}_q$.

**Lemma 3** ([27])**.** *The quadratic Wasserstein distance between* $\mathcal{N}(\boldsymbol{\mu}_1, \boldsymbol{\Sigma}_1)$ *and* $\mathcal{N}(\boldsymbol{\mu}_2, \boldsymbol{\Sigma}_2)$

$$\sqrt{\|\boldsymbol{\mu}_1 - \boldsymbol{\mu}_2\|_2^2 + \mathrm{tr}(\boldsymbol{\Sigma}_1 + \boldsymbol{\Sigma}_2 - 2(\Sigma_1^{\frac{1}{2}}\boldsymbol{\Sigma}_2\boldsymbol{\Sigma}_1^{\frac{1}{2}})^{\frac{1}{2}})}. \tag{3}$$

The lemma above indicates that the quadratic Wasserstein distance can be easily computed using the population means and covariance matrices. In practice, we estimate these population quantities, $\boldsymbol{\mu}_1, \boldsymbol{\mu}_2, \boldsymbol{\Sigma}_1$, and $\boldsymbol{\Sigma}_2$, with their sample counterparts: $\widehat{\boldsymbol{\mu}}_1, \widehat{\boldsymbol{\mu}}_2, \widehat{\boldsymbol{\Sigma}}_1$, and $\widehat{\boldsymbol{\Sigma}}_2$. The empirical quadratic Wasserstein distance is then used as the optimization objective to align the feature and codebook distributions:

$$\mathcal{L}_{\mathcal{W}} = \sqrt{\|\widehat{\boldsymbol{\mu}}_1 - \widehat{\boldsymbol{\mu}}_2\|_2^2 + \mathrm{tr}(\widehat{\boldsymbol{\Sigma}}_1 + \widehat{\boldsymbol{\Sigma}}_2 - 2(\widehat{\boldsymbol{\Sigma}}_1^{\frac{1}{2}}\widehat{\boldsymbol{\Sigma}}_2\widehat{\boldsymbol{\Sigma}}_1^{\frac{1}{2}})^{\frac{1}{2}})}. \tag{4}$$

A smaller value of $\mathcal{L}_{\mathcal{W}}$ indicates stronger alignment between the feature distribution $\mathcal{P}_A$ and the codebook distribution $\mathcal{P}_B$. We refer to the VQ algorithm that employs $\mathcal{L}_{\mathcal{W}}$ as *Wasserstein VQ*.

## 3.2 Integration into the VQ-VAE Framework

We first examine *Wasserstein VQ* within the VQ-VAE framework [36]. As illustrated in the Figure 4, the VQ-VAE model combines three key components: an encoder $E(\cdot)$, a decoder $D(\cdot)$, a quantizer $\mathcal{Q}(\cdot)$ with a learnable codebook $\{\mathbf{e}_k\}_{k=1}^K$. As described earlier in Section 2.1, for an input image $\boldsymbol{x}$, the encoder processes the image to yield a spatial feature $\boldsymbol{z}_e = E(\boldsymbol{x}) \in \mathbb{R}^{h \times w \times d}$. The quantizer converts $\boldsymbol{z}_e$ into a quantized feature $\boldsymbol{z}_q$, from which the decoder reconstructs the image as $\widehat{\boldsymbol{x}} = D(\boldsymbol{z}_q)$. By incorporating our proposed Wasserstein loss $\mathcal{L}_{\mathcal{W}}$ into the VQ-VAE framework, the overall loss objective can be formulated as follows:

$$\mathcal{L}_{\text{VQ-VAE}} = \|\widehat{\boldsymbol{x}} - \boldsymbol{x}\|_2^2 + \beta\|\mathrm{sg}(\boldsymbol{z}_q) - \boldsymbol{z}_e\|_2^2 \tag{5}$$
$$+ \|\mathrm{sg}(\boldsymbol{z}_e) - \boldsymbol{z}_q\|_2^2 + \gamma\mathcal{L}_{\mathcal{W}}.$$

where sg denotes the stop-gradient operation. $\beta$ and $\gamma$ are hyper-parameters. We set $\gamma = 0.5$ for all experiments.

## 3.3 Integration into the VQGAN Framework

To ensure high perceptual quality in the reconstructed images, we further investigate *Wasserstein VQ* within the VQGAN framework [9]. VQGAN extends the VQ-VAE framework by integrating a VGG network [32] and a patch-based discriminator [9, 15]. The overall training objective of VQGAN can be written as follows:

$$\mathcal{L}_{\text{VQGAN}} = \mathcal{L}_{\text{VQ-VAE}} + \mathcal{L}_{\text{Per}} + \lambda\mathcal{L}_{\text{GAN}}. \tag{6}$$

Where $\mathcal{L}_{\text{Per}}$ and $\mathcal{L}_{\text{GAN}}$ denote the VGG-based perceptual loss [41], and GAN loss [14, 21], respectively. We set $\lambda = 0.2$ for all experiments.

Table 1: Comparison of VQ-VAEs trained on FFHQ dataset following [36].

| Approach | Tokens | Codebook Size | $\mathcal{U}$ (↑) | $\mathcal{C}$ (↑) | PSNR(↑) | SSIM(↑) | Rec. Loss (↓) |
|---|---|---|---|---|---|---|---|
| Vanilla VQ | 256 | 16384 | 3.8% | 527.2 | 27.83 | 73.8 | 0.0119 |
| EMA VQ | 256 | 16384 | 14.0% | 1795.7 | 28.39 | 74.8 | 0.0106 |
| Online VQ | 256 | 16384 | 11.7% | 1115.3 | 27.68 | 72.6 | 0.0125 |
| **Wasserstein VQ** | 256 | 16384 | **100%** | **15713.3** | **29.03** | **76.6** | **0.0093** |
| Vanilla VQ | 256 | 50000 | 1.2% | 516.8 | 27.83 | 73.6 | 0.0120 |
| EMA VQ | 256 | 50000 | 10.3% | 4075.7 | 28.61 | 75.3 | 0.0101 |
| Online VQ | 256 | 50000 | 6.0% | 1642.9 | 28.37 | 74.6 | 0.0107 |
| **Wasserstein VQ** | 256 | 50000 | **100%** | **47496.4** | **29.24** | **77.0** | **0.0089** |
| Vanilla VQ | 256 | 100000 | 0.6% | 481.0 | 27.86 | 74.2 | 0.0118 |
| EMA VQ | 256 | 100000 | 2.7% | 2087.5 | 28.43 | 74.8 | 0.0105 |
| Online VQ | 256 | 100000 | 3.6% | 1556.8 | 27.12 | 71.1 | 0.0142 |
| **Wasserstein VQ** | 256 | 100000 | **100%** | **93152.7** | **29.53** | **78.0** | **0.0083** |

Table 2: Comparison of VQ-VAEs trained on ImageNet dataset following [36].

| Approach | Tokens | Codebook Size | $\mathcal{U}$ (↑) | $\mathcal{C}$ (↑) | PSNR(↑) | SSIM(↑) | Rec. Loss (↓) |
|---|---|---|---|---|---|---|---|
| Vanilla VQ | 256 | 16384 | 2.5% | 360.7 | 24.44 | 57.5 | 0.0294 |
| EMA VQ | 256 | 16384 | 14.5% | 1861.5 | 24.98 | 59.2 | 0.0267 |
| Online VQ | 256 | 16384 | 22.2% | 1465.6 | 24.88 | 58.6 | 0.0273 |
| **Wasserstein VQ** | 256 | 16384 | **100%** | **15539.1** | **25.47** | **61.2** | **0.0242** |
| Vanilla VQ | 256 | 50000 | 0.9% | 378.7 | 24.40 | 57.7 | 0.0295 |
| EMA VQ | 256 | 50000 | 16.8% | 6139.3 | 25.37 | 60.9 | 0.0246 |
| Online VQ | 256 | 50000 | 9.9% | 2241.7 | 25.09 | 59.7 | 0.0260 |
| **Wasserstein VQ** | 256 | 50000 | **100%** | **46133.2** | **25.72** | **62.3** | **0.0230** |
| Vanilla VQ | 256 | 100000 | 0.4% | 337.0 | 24.43 | 57.4 | 0.0295 |
| EMA VQ | 256 | 100000 | 3.0% | 2170.0 | 25.13 | 60.1 | 0.0257 |
| Online VQ | 256 | 100000 | 4.1% | 1709.9 | 24.95 | 59.1 | 0.0267 |
| **Wasserstein VQ** | 256 | 100000 | **100%** | **93264.7** | **25.88** | **63.0** | **0.0223** |

## 4 Experiments

In this section, we empirically demonstrate the effectiveness of our proposed *Wasserstein VQ* algorithm in visual tokenization tasks. Our experiments are conducted within the frameworks of VQ-VAE [36] and VQGAN [9]. The PyTorch code, including training environment, scripts and logs, will be made publicly available.

### 4.1 Evaluation on VQ-VAE Framework

**Datasets and Baselines** Experiments are conducted on four benchmark datasets: two low-resolution datasets, i.e., CIFAR-10 [18] and SVHN [26], and two high-resolution datasets FFHQ [16] and ImageNet [7]. We evaluated our approach against several representative VQ methods: Vanilla VQ [36], EMA VQ [29], which uses exponential moving average updates and is also referred to as $k$-means, Online VQ, which employs $k$-means++ in CVQ-VAE [42]. For detailed experimental settings, please refer to Appendix J.

**Metrics** We employ multiple evaluation metrics, including the Codebook Utilization Rate ($\mathcal{U}$), Codebook Perplexity ($\mathcal{C}$), peak signal-to-noise ratio (PSNR), patch-level structural similarity index (SSIM), and pixel-level reconstruction loss (Rec. Loss). We exclude the quantization error ($\mathcal{E}$) from our reported results, as it is highly sensitive to distribution variances—a factor analyzed in Appendix G. Since these distribution variances remain uncontrolled in our experiments, fair comparison based on ($\mathcal{E}$) would be unreliable. To ensure an equitable assessment, Appendix H provides an atomic setting where distribution variances are fully controlled and identical across all VQ variants.

**Main Results** As shown in Tables 1, 2, and Tables 6, 7 in the Appendix K, our proposed *Wasserstein VQ* outperforms all baselines on both datasets, achieving superior performance across almost all evaluation metrics under various experimental settings. The underlying reason is that VQ inherently functions as a compressor, transitioning from a continuous latent space to a discrete space, where minimal information loss indicates improved expressivity. Our proposed *Wasserstein VQ* employs explicit distribution matching constraints, thereby achieving a more favorable alignment between

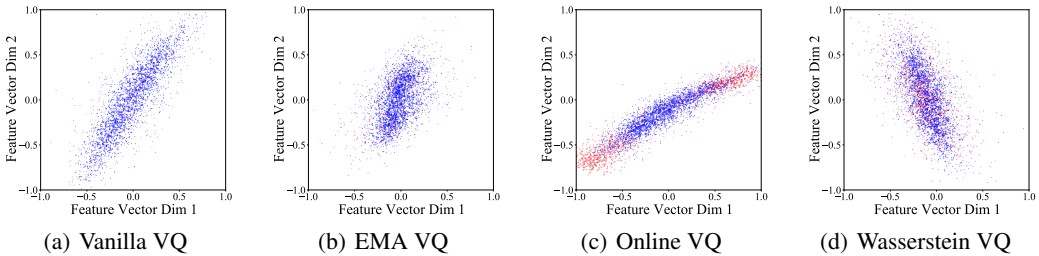

| (a) Vanilla VQ | (b) EMA VQ | (c) Online VQ | (d) Wasserstein VQ |

Figure 5: Visualization of feature and codebook distributions. The symbols blue · and red × represent the feature and code vectors, respectively.

the feature vectors and code vectors. This results in nearly 100% codebook utilization and almost minimal quantization error, leading to the lowest Rec. Loss among all settings.

**Representation Visualization**   To visualize the distributions of feature vectors and code vectors across different VQ methods trained on the FFHQ dataset (with a fixed codebook size of $8192$), we randomly sample 3000 feature vectors and 1000 code vectors and plot their scatter diagrams. As shown in Figure 5(a) and Figure 5(b), in Vanilla VQ and EMA VQ, the majority of code vectors are clustered near the zero point, rendering them effectively unusable. While Online VQ avoids this central clustering issue, most of its code vectors are distributed at the two extremes of the feature space, as illustrated in Figure 5(c). This distributional mismatch leads to increased information loss and reduced codebook utilization. In contrast to these three VQ methods, *Wasserstein VQ* demonstrates significantly better distributional matching between feature vectors and code vectors. This alignment substantially minimizes information loss and enhances codebook utilization.

**Gaussian Hypothesis Justification**   To justify the reasonableness of the Gaussian assumption, we extract feature vectors from the encoder and computed the density of arbitrary two dimensions by binning the data points into 29 groups, as visualized in Figure 6(a). Furthermore, we randomly selected 2000 data points from any two dimensions and plotted them in a scatter plot, as shown in Figure 6(b). It is evident that the feature vectors exhibit Gaussian-like characteristics. The under-

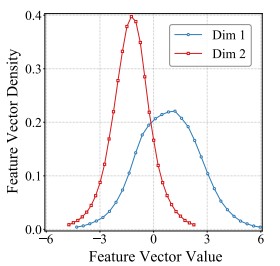
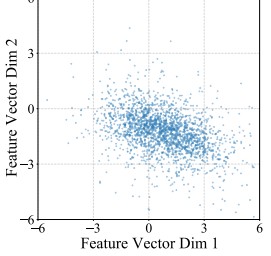

| (a) Feature Density | (b) Feature Visualization |

Figure 6: Visualization of feature vectors.

lying reason for this behavior can be attributed to the central limit theorem, which posits that learned feature vectors and code vectors will approximate a Gaussian distribution given a sufficiently large sample size and a relatively low-dimensional space, i.e., $d = 8$.

**Analyses of Codebook Size**   We investigate the impact of the codebook size $K$ on VQ performance, as presented in Table 1, and Table 8 in Appendix L. Vanilla VQ suffers from severe codebook collapse even with a small $K$, such as $K = 1024$. In contrast, improved algorithms, such as EMA VQ and Online VQ, also experience codebook collapse when $K$ is very large, e.g., $K \geq 50000$. Notably, *Wasserstein VQ* consistently maintains 100% codebook utilization, regardless of the codebook size. This demonstrates that distributional matching by quadratic Wasserstein distance effectively resolves the issue of codebook collapse.

**Analyses of Codebook Dimensionality**   We further investigate the impact of codebook dimensionality $d$ on VQ performance. We conduct experiments on CIFAR-10 dataset and range $d$ from 2 to 32. As shown in Table 9 in Appendix L our proposed Wasserstein VQ consistently outperforms all baselines regardless of dimensionality. Notably, we observe the curse of dimensionality phenomenon—performance degrades as dimensionality increases. Vanilla VQ exhibits the most severe degradation, followed by EMA VQ and Online VQ, while our Wasserstein VQ shows only minimal codebook utilization reduction.

Table 3: Comparison of VQGAN trained on FFHQ dataset following [9].

| Method | Tokens | Codebook Size | Utilization (%) ↑ | rFID ↓ | LPIPS ↓ | PSNR ↑ | SSIM ↑ |
|---|---|---|---|---|---|---|---|
| RQVAE[†] [19] | 256 | 2,048 | - | 7.04 | 0.13 | 22.9 | 67.0 |
| VQ-WAE[†] [37] | 256 | 1,024 | - | 4.20 | 0.12 | 22.5 | 66.5 |
| MQVAE[†] [13] | 256 | 1,024 | 78.2 | 4.55 | - | - | - |
| VQGAN[†] [9] | 256 | 16,384 | 2.3 | 5.25 | 0.12 | 24.4 | 63.3 |
| VQGAN-FC[†] [39] | 256 | 16,384 | 10.9 | 4.86 | 0.11 | 24.8 | 64.6 |
| VQGAN-EMA[†] [29] | 256 | 16,384 | 68.2 | 4.79 | 0.10 | 25.4 | 66.1 |
| VQGAN-LC[†] [43] | 256 | 100,000 | 99.5 | 3.81 | 0.08 | 26.1 | 69.4 |
| | 256 | 16,384 | 100 | 3.08 | 0.08 | 26.3 | 70.4 |
| Wasserstein VQ[⋆] | 256 | 50,000 | 100 | 2.96 | 0.08 | 26.5 | 71.4 |
| | 256 | 100,000 | 100 | **2.71** | **0.07** | **26.6** | **71.9** |
| | 680 | 16,384 | 100 | 2.48 | 0.06 | 27.4 | 74.0 |
| Multi-scale Wasserstein VQ[⋆] | 680 | 50,000 | 100 | 2.07 | 0.06 | 27.6 | 74.6 |
| | 680 | 100,000 | 100 | **1.79** | **0.05** | **27.9** | **75.4** |

## 4.2 Evaluation on VQGAN Framework

**Dataset, Baselines, and Metrics**   We evaluated our approach against following methods on the FFHQ dataset: RQVAE [19], VQGAN [9], VQGAN-FC [39], VQGAN-EMA [29], VQ-WAE [37], MQVAE [13], and VQGAN-LC [43]. Following VQGAN-LC [43], we employ the Fréchet Inception Distance (r-FID) [11], Learned Perceptual Image Patch Similarity (LPIPS) [41], PSNR, and SSIM to evaluate visual reconstruction quality.

**Main Results**   As presented in Table 3, our proposed *Wasserstein VQ* outperforms all baselines across all evaluation metrics within the VQGAN framework. This superior performance stems from its VQ system that minimizes information loss, as discussed in Section 4.1, thereby achieving optimal reconstruction fidelity and visual perceptual quality. Notably, when integrating VAR's multi-scale VQ [35] with our *Wasserstein VQ*, we observe a significant improvement in rFID (reduced from 2.71 to 1.79 with codebook size $K = 100000$). Figure 7 demonstrates that *Wasserstein VQ*'s reconstructed images exhibit only minimal differences from the inputs, confirming its exceptional visual tokenization capability.

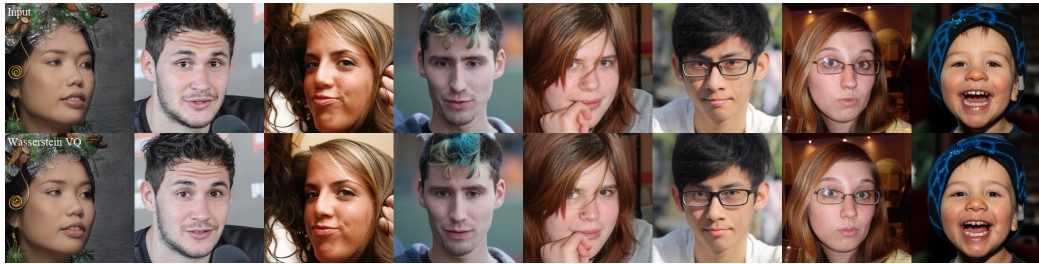

Figure 7: Visualization of reconstructed Images. The top row displays the original input images with a resolution of $256 \times 256$ pixels, while the bottom row shows the reconstructed images from the *Wasserstein VQ*.

## 5 Conclusion

This paper examines vector quantization (VQ) from a distributional perspective, introducing three key evaluation criteria. Empirical results demonstrate that optimal VQ results are achieved when the distributions of continuous feature vectors and code vectors are identical. Our theoretical analysis confirms this finding, emphasizing the crucial role of distributional alignment in effective VQ. Based on these insights, we propose using the quadratic Wasserstein distance to achieve alignment, leveraging its computational efficiency under a Gaussian hypothesis. This approach achieves near-full codebook utilization while significantly reducing quantization error. Our method successfully addresses both training instability and codebook collapse, leading to improved downstream image reconstruction performance. A limitation of this work, however, is that our proposed distributional matching approach relies on the assumption of Gaussian distribution, which may not strictly hold in all scenarios. In future work, we aim to develop methods that do not depend on this assumption, thereby broadening the applicability and robustness of our VQ framework.

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

# A Optimal Support of The Codebook Distribution

*Proof of Theorem 1.* First, we assume $\overline{\text{supp}(\mathcal{P}_B)} = \overline{\text{supp}(\mathcal{P}_A)}$. Then for any $z \in \text{supp}(\mathcal{P}_A)$, there exist a sequence of points in $\text{supp}(\mathcal{P}_B)$ that converge to $z$. Let $\{e_k\}_{k=1}^K$ be $K$ code vectors independently generated from $\mathcal{P}_B$. Then the empirical distribution of $\{e_k\}_{k=1}^K$ tends to $\mathcal{P}_B$ as the size $K$ tends to infinity. Since $\Omega = \text{supp}(\mathcal{P}_A)$ is a bounded region, we have the following:

$$\sup_{z \in \overline{\text{supp}(\mathcal{P}_A)}} \min_k \|z - e_k\|^2 = \sup_{z \in \overline{\text{supp}(\mathcal{P}_B)}} \min_k \|z - e_k\|^2 \xrightarrow{P} 0, \quad \text{as } K \to \infty.$$

This quantity is an upper bound on the quantization error $\mathcal{E}(\{z_i\}; \{e_k\})$. Thus,

$$\sup_{\{z_i\} \subseteq \Omega} \mathcal{E}\left(\{z_i\}_{i=1}^N; \{e_k\}_{k=1}^K\right) \leq \sup_{z \in \overline{\Omega}} \min_k \|z - e_k\|^2 \xrightarrow{P} 0, \quad \text{as } K \to \infty.$$

This demonstrates that $\mathcal{P}_B$ has vanishing quantization error asymptotically. Furthermore, for any $K$ code vectors $\{e_k\}_{k=1}^K$ independently drawn from $\mathcal{P}_B$, we have $\{e_k\}_{k=1}^K \subseteq \overline{\Omega}$. Since the empirical distribution of $\{z_i\}_{i=1}^N$ tends to $\mathcal{P}_A$ as the feature sample size $N$ tends to infinity, we can easily show that for any fixed $\{e_k\}_{k=1}^K \subseteq \overline{\Omega}$, the codebook utility rate satisfies

$$\mathcal{U}\left(\{z_i\}_{i=1}^N, \{e_k\}_{k=1}^K\right) \xrightarrow{p} 1, \quad \text{as } N \to \infty.$$

This shows that $\{e_k\}_{k=1}^K$ attains full utilization asymptotically, and thus $\mathcal{P}_B$ attains full utilization asymptotically.

On the other hand, we assume $\mathcal{P}_B$ attains full utilization and vanishing quantization error asymptotically. Then we first claim that $\text{supp}(\mathcal{P}_A) \subseteq \overline{\text{supp}(\mathcal{P}_B)}$. Since $\mathcal{P}_B$ has vanishing quantization error asymptotically, then for any $z \in \text{supp}(\mathcal{P}_A)$, there exist a sequence of points in $\text{supp}(\mathcal{P}_B)$ that converge to $z$. This implies that $\text{supp}(\mathcal{P}_A) \subseteq \overline{\text{supp}(\mathcal{P}_B)}$ and thus $\overline{\text{supp}(\mathcal{P}_A)} \subseteq \overline{\text{supp}(\mathcal{P}_B)}$.

To show $\overline{\text{supp}(\mathcal{P}_B)} = \overline{\text{supp}(\mathcal{P}_A)}$, it remains to show $\text{supp}(\mathcal{P}_B) \subseteq \overline{\text{supp}(\mathcal{P}_A)}$. In fact, if $\text{supp}(\mathcal{P}_B) \subseteq \overline{\text{supp}(\mathcal{P}_A)}$ does not hold, then there exists an open region $\mathcal{R} \subseteq \text{supp}(\mathcal{P}_B) - \overline{\text{supp}(\mathcal{P}_A)}$ such that $\mathcal{P}_B(\mathcal{R}) > 0$ and

$$\min_{z \in \text{supp}(\mathcal{P}_A), z' \in \mathcal{R}} \|z - z'\| \geq \epsilon_0$$

for some $\epsilon_0 > 0$. Since $\text{supp}(\mathcal{P}_A) \subseteq \overline{\text{supp}(\mathcal{P}_B)}$, then there exists a sufficiently large $K_0$ such that the event

$$\left\{ \text{Generating}\{e_k\}_{k=1}^{K_0} \text{ i.i.d. from } \mathcal{P}_B \text{ s.t. } \{e_k\} \subseteq \text{supp}(\mathcal{P}_A), \sup_{z \in \text{supp}(\mathcal{P}_A)} \min_k \|z - e_k\| < \epsilon_0 \right\} \quad (7)$$

has some positive probability $C > 0$. Then with a positive probability of at least $C \cdot \mathcal{P}_B(\mathcal{R})$, we can pick the first $K_0$ code vectors from Equation (7) and the $(K_0 + 1)$th code vector from $\mathcal{R}$. For any such codebook of size $K_0 + 1$, we know the $(K_0 + 1)$th code vector will never be used regardless of the choice of the feature set $\{z_i\}$. Therefore, the codebook utilization

$$\sup_{\{z_i\}} \mathcal{U}\left(\{e_k\}_{k=1}^{K_0+1}; \{z_i\}\right) \leq \frac{K_0}{K_0 + 1} < 1.$$

This contradicts the property that $\mathcal{P}_B$ attains full utilization asymptotically. Thus, $\text{supp}(\mathcal{P}_B) \subseteq \overline{\text{supp}(\mathcal{P}_A)}$ must hold. This concludes the proof. $\qquad\square$

# B Statistical Distances over Gaussian Distributions

We first introduce the definition of Wasserstein distance.

**Definition 4.** *The Wasserstein distance or earth-mover distance with p norm is defined as below:*

$$W_p(\mathbb{P}_r, \mathbb{P}_g) = \left(\inf_{\gamma \in \Pi(\mathbb{P}_r, \mathbb{P}_g)} \mathbb{E}_{(x,y) \sim \gamma}\left[\|x - y\|^p\right]\right)^{1/p}. \quad (8)$$

where $\Pi(\mathcal{P}_r, \mathcal{P}_g)$ denotes the set of all joint distributions $\gamma(x, y)$ whose marginals are $\mathcal{P}_r$ and $\mathcal{P}_g$ respectively. Intuitively, when viewing each distribution as a unit amount of earth/soil, the Wasserstein distance (also known as earth-mover distance) represents the minimum cost of transporting "mass" from $x$ to $y$ to transform distribution $\mathcal{P}_r$ into distribution $\mathcal{P}_g$. When $p = 2$, this is called the quadratic Wasserstein distance.

In this paper, we achieve distributional matching using the quadratic Wasserstein distance under Gaussian distribution assumptions. We also examine other statistical distribution distances as potential loss functions for distributional matching and compare them with the Wasserstein distance. Specifically, we provide the Kullback-Leibler divergence and the Bhattacharyya distance over Gaussian distributions in Lemma 5 and Lemma 6. It can be observed that the KL divergence for two Gaussian distributions involves calculating the determinant of covariance matrices, which is computationally expensive in moderate and high dimensions. Moreover, the calculation of the determinant is sensitive to perturbations and it requires full rank (In the case of not full rank, the determinant is zero, rendering the logarithm of zero undefined), which can be impractical in many cases. Other statistical distances like Bhattacharyya Distance suffer from the same issue. In contrast, quadratic Wasserstein distance does not require the calculation of the determinant and full-rank covariance matrices.

**Lemma 5** (Kullback-Leibler divergence [22]). *Suppose two random variables $\mathbf{Z}_1 \sim \mathcal{N}(\boldsymbol{\mu}_1, \boldsymbol{\Sigma}_1)$ and $\mathbf{Z}_2 \sim \mathcal{N}(\boldsymbol{\mu}_2, \boldsymbol{\Sigma}_2)$ obey multivariate normal distributions, then Kullback-Leibler divergence between $\mathbf{Z}1$ and $\mathbf{Z}_2$ is:*

$$D_{\mathrm{KL}}(\mathbf{Z}_1, \mathbf{Z}_2) = \frac{1}{2}((\boldsymbol{\mu}_1 - \boldsymbol{\mu}_2)^T \boldsymbol{\Sigma}_2^{-1}(\boldsymbol{\mu}_1 - \boldsymbol{\mu}_2) + \mathrm{tr}(\boldsymbol{\Sigma}_2^{-1}\boldsymbol{\Sigma}_1 - \mathbf{I}) + \ln \frac{\det \boldsymbol{\Sigma}_2}{\det \boldsymbol{\Sigma}_1}).$$

**Lemma 6** (Bhattacharyya Distance [4]). *Suppose two random variables $\mathbf{Z}_1 \sim \mathcal{N}(\boldsymbol{\mu}_1, \boldsymbol{\Sigma}_1)$ and $\mathbf{Z}_2 \sim \mathcal{N}(\boldsymbol{\mu}_2, \boldsymbol{\Sigma}_2)$ obey multivariate normal distributions, $\boldsymbol{\Sigma} = \frac{1}{2}(\boldsymbol{\Sigma}_1 + \boldsymbol{\Sigma}_2)$, then bhattacharyya distance between $\mathbf{Z}1$ and $\mathbf{Z}_2$ is:*

$$\mathcal{D}_B(\mathbf{Z}_1, \mathbf{Z}_2) = \frac{1}{8}(\boldsymbol{\mu}_1 - \boldsymbol{\mu}_2)^T \boldsymbol{\Sigma}^{-1}(\boldsymbol{\mu}_1 - \boldsymbol{\mu}_2) + \frac{1}{2} \ln \frac{\det \boldsymbol{\Sigma}}{\sqrt{\det \boldsymbol{\Sigma}_1 \det \boldsymbol{\Sigma}_2}}.$$

# C  Understanding Codebook Collapse Through the Lens of Voronoi Partition

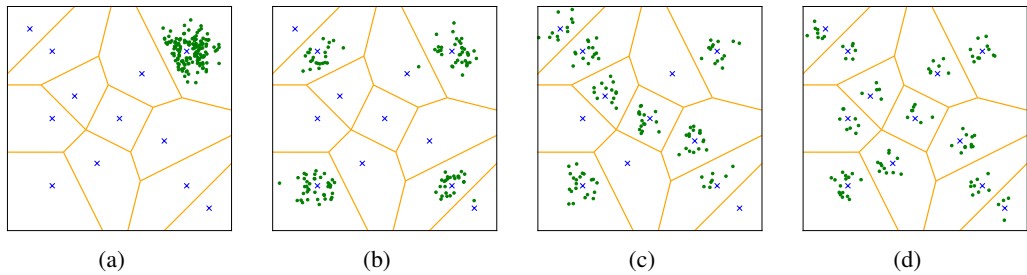

Figure 8: Visualization of the Voronoi partition. The symbols $\cdot$ and $\times$ represent the feature and code vectors, respectively.

## C.1  The Definition of Voronoi Partition and Its Connection to Codebook Collapse

Let $\mathcal{X}$ be a metric space with distance function $d(\cdot, \cdot)$, and given a set of code vectors $\{\boldsymbol{e}_k\}_{k=1}^K$. The Voronoi cell, or Voronoi region, $\mathcal{R}_k$, associated with the code vector $\boldsymbol{e}_k$ is the set of all points in $\mathcal{X}$ whose distance to $\boldsymbol{e}_k$ is not greater than their distance to the other code vectors $\boldsymbol{e}_j$, where $j$ is any index different from $k$. Mathematically, this can be expressed as:

$$\mathcal{R}_k = \{x \in \mathcal{X}; d(x, \boldsymbol{e}_k) \leq d(x, \boldsymbol{e}_j), \forall j \neq k\}, \tag{9}$$

The Voronoi diagram is simply the tuple of cells $\{\mathcal{R}_k\}_{k=1}^K$. As depicted in Figure 8, there are 12 code vectors which partition the metric space into 12 regions according to the $\mathcal{R}_k$. When $d$ is a a distance function based on the $\ell_2$ norm, the vector quantization (VQ) process can be equivalently understood through the regions $\mathcal{R}_k$ as:

$$\forall \boldsymbol{z}_i \in \mathcal{R}_j, \quad \boldsymbol{z}_i' = \underset{\boldsymbol{e} \in \{\boldsymbol{e}_k\}}{\arg\min} \|\boldsymbol{z}_i - \boldsymbol{e}\| = \boldsymbol{e}_j \tag{10}$$

Where $\boldsymbol{z}_i$ is an arbitrary feature vector. Equation 10 offers an alternative approach for nearest neighbor search in code vector selection. Specifically, this involves first identifying the partition region $\mathcal{R}_j$ to which the feature vector $\boldsymbol{z}_i$ belongs, and then directly obtaining the nearest code vector $\boldsymbol{e}_j$ based on the region's id $j$.

**Relation to Codebook Collapse** The most severe case of codebook collapse occurs when all feature vectors belong to the same partition region. As illustrated in Figure 8(a), all feature vectors are confined to a single partition region in the upper right corner, resulting in the utilization of only one code vector. To prevent codebook collapse, it is crucial for feature vectors to be distributed across all partition regions as evenly as possible, as depicted in Figure 8(d).

## C.2 Why Existing Vector Quantization Strategies Fail to Address Codebook Collapse

This section offers an in-depth analysis of why existing VQ methods inherently struggle to address codebook collapse. We use Vanilla VQ [36] and VQ methods based on the $k$-means algorithm [29] as illustrative examples. These two approaches share similarities in their assignment step but differ in their update mechanisms.

**Assignment Step** Suppose there is a set of feature vectors $\{\boldsymbol{z}_i\}_{i=1}^N$ and code vectors $\{\boldsymbol{e}_k\}_{k=1}^K$. In the $t$-th assignment step, both algorithms partition the feature space into Voronoi cells, based on which we assign all feature vectors to their nearest code vectors as follows:

$$\mathcal{R}_k^{(t-1)} = \{x \in \mathcal{X}; \left\|x - \boldsymbol{e}_k^{(t-1)}\right\|_2^2 \leq \left\|x - \boldsymbol{e}_j^{(t-1)}\right\|_2^2, \forall j \neq k\}, \quad \mathcal{S}_k^{(t)} = \{\boldsymbol{z}_i; \boldsymbol{z}_i \in \mathcal{R}_k^{(t-1)}\} \quad (11)$$

**Update Step in Vanilla VQ** It updates the code vectors using gradient descent through the loss function provided below.

$$\mathcal{L} = \frac{1}{N} \sum_{k=1}^K \sum_{z_m \in \mathcal{S}_k^{(t)}} \left\|\boldsymbol{z}_m - \boldsymbol{e}_k^{(t-1)}\right\|_2^2 \quad (12)$$

**Update Step in $k$-means-based VQ** It updates the code vectors by using an exponential moving average of the feature vectors assigned to each code vector:

$$\boldsymbol{e}_k^{(t)} = \alpha \boldsymbol{e}_k^{(t-1)} + (1 - \alpha)\frac{1}{|\mathcal{S}_k^{(t)}|} \sum_{\boldsymbol{z}_m \in \mathcal{S}_k^{(t)}} \boldsymbol{z}_m \quad (13)$$

**Codebook Collapse in Two VQs** While both VQ methods employ different update strategies for the code vectors, they still suffer from codebook collapse. This is because, in the assignment step, the learnable Voronoi partition does not guarantee that all Voronoi cells will be assigned feature vectors, as illustrated in Figures 8(a) to 8(c). Especially when the codebook size is large, there are more Voronoi cells, and inevitably, some cells remain unassigned. In such cases, the corresponding code vectors remain unupdated and underutilized.

**Connection to Distribution Matching and Solutions** In Appendix H, we demonstrated through synthetic experiments that the effectiveness of both VQ methods heavily relies on the codebook initialization. Only when the codebook distribution is initialized to approximate the feature distribution can codebook collapse be effectively mitigated. However, in practical applications, the feature distribution is often unknown and evolves dynamically during training. To address this issue, we propose an explicit distributional matching constraint that ensures the codebook distribution aligns closely with the feature distribution, thereby achieving 100% codebook utilization.

## D Complementary Roles of Criterion 2 and 3 in Assessing Codebook Collapse

To explain the complementary roles of Criterion 2 and 3 (defined in Section 2.2), we provide visual elucidations for enhanced clarity and understanding. The metric $\mathcal{U}$ is capable of quantifying the *completeness* of codebook utilization. As depicted in the Figure 9(a) and 9(b), the values of $\mathcal{U}$ are

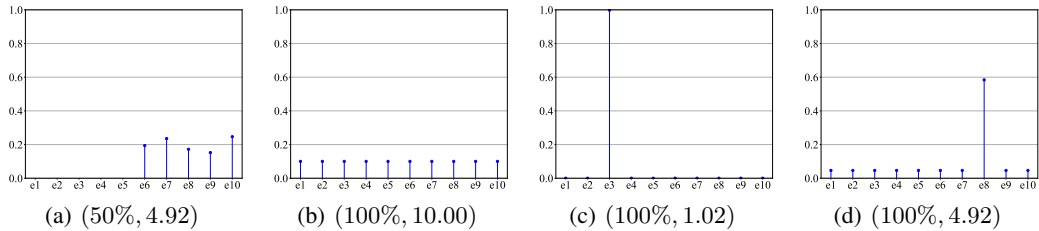

|  |  |  |  |
|---|---|---|---|
| (a) $(50\%, 4.92)$ | (b) $(100\%, 10.00)$ | (c) $(100\%, 1.02)$ | (d) $(100\%, 4.92)$ |

Figure 9: Visualization of the evaluation criteria $(\mathcal{U}, \mathcal{C})$.

$50\%$ and $100\%$, respectively[2]. However, $\mathcal{U}$ alone is insufficient to evaluate the degree of codebook collapse, as it fails to address the scenario depicted in Figure 9(c). Although all code vectors are utilized, the code vector $e_3$ excessively dominates the codebook utilization, resulting in an extreme imbalance. This imbalanced codebook utilization can be considered a form of codebook collapse, despite $\mathcal{U}$ reaching its maximum value. This observation motivates the proposal of Criterion 3, which is capable of gauging the *imbalance* or *uniformity* inherent in codebook utilization.

When compared in Figure 9(b) and 9(c), the value of $\mathcal{C}$ are $10.00$ and $1.02$, respectively, demonstrating that Criterion 3 is capable of distinguishing the imbalance of code vector utilization $p_k$ under conditions where cases share the same $\mathcal{U}$, e.g., $\mathcal{U} = 100\%$. Additionally, Criterion 3 categorizes Figure 9(c) as indicative of codebook collapse, as the value $\mathcal{C}$ nearly reaches its minimum of $1.0$, a result that resonates with our desired interpretation. However, it is essential to note that Criterion 3 alone does not suffice to evaluate the degree of codebook collapse. When scrutinizing Figure 9(a) and 9(d), despite the identical $\mathcal{C}$, there exists a stark disparity in $\mathcal{U}$. This observation underscores that the value of $\mathcal{C}$ is inadequate for quantifying the proportion of actively utilized code vectors.

In this paper, we adopt the combination of Criterion 2 and 3 to quantitatively assess the extent of codebook collapse. A robust mitigation of codebook collapse is indicated solely when both $\mathcal{U}$ and $\mathcal{C}$ exhibit substantial values.

# E    Interpretation of Qualitative Distributional Matching Results in Figure 3

This section interprets the experimental results presented in Figure 3. The VQ process relies on nearest neighbor search for code vector selection. As evident from Figure 3(a) to 3(d), actively selected code vectors are predominantly those located in close proximity to or within the feature distribution, while distant ones remain unselected. This leads to highly uneven code vector utilization $p_k$, with those closer to the feature distribution being excessively used. This elucidates the significantly low $\mathcal{U}$ and $\mathcal{C}$ observed in Figure 3(a). Furthermore, a notable quantization error, e.g., $\mathcal{E} = 1.19$ in Figure 3(a), arises when the codebook and feature distributions are mismatched, forcing feature vectors outside the codebook to settle for distant code vectors. Conversely, as the disk centers align, leading to a closer match between the two distributions, an increased number of code vectors become actively engaged. Additionally, code vectors are utilized more uniformly, and feature vectors can select nearer counterparts. This accounts for the improvement of criterion triple values towards optimality as the distributions align.

Analogously, we can employ nearest neighbor search to interpret the second case. When code vectors are distributed within the range of feature vectors, as illustrated in Figure 3(e) and Figure 3(f), the majority of code vectors would be actively utilized, ensuring high $\mathcal{U}$. However, the utilization of these code vectors is not uniform; code vectors on the periphery of the codebook distribution are more frequently used, leading to relatively low $\mathcal{C}$. Feature vectors on the periphery will have larger distances to their nearest code vectors, resulting in higher $\mathcal{E}$. Conversely, when feature vectors fall within the range of code vectors, as depicted in Figure 3(g) and Figure 3(h), outer code vectors remain largely unused, leading to a lower $\mathcal{U}$ and $\mathcal{C}$. Since only inner code vectors are active, each feature vector can find a nearby counterpart, maintaining low $\mathcal{E}$.

---

[2]This discrepancy arises because, in Figure 9(a) only half of code vectors' utilization $p_k$ (as defined in Criterion 3) exceeds zero, whereas in Figure 9(b), the utilization $p_k$ of of all code vectors surpasses zero.

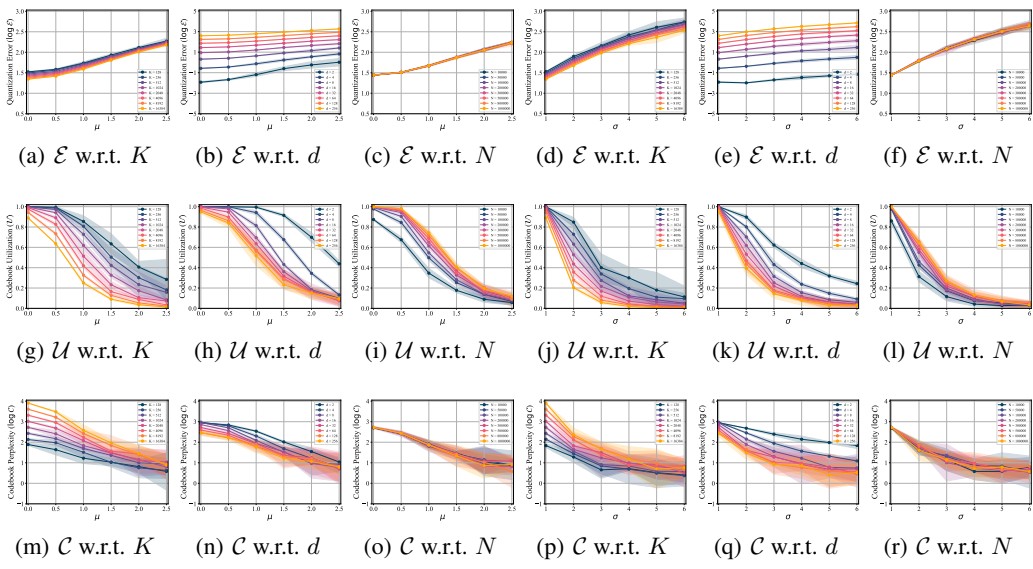

| (a) $\mathcal{E}$ w.r.t. $K$ | (b) $\mathcal{E}$ w.r.t. $d$ | (c) $\mathcal{E}$ w.r.t. $N$ | (d) $\mathcal{E}$ w.r.t. $K$ | (e) $\mathcal{E}$ w.r.t. $d$ | (f) $\mathcal{E}$ w.r.t. $N$ |

| (g) $\mathcal{U}$ w.r.t. $K$ | (h) $\mathcal{U}$ w.r.t. $d$ | (i) $\mathcal{U}$ w.r.t. $N$ | (j) $\mathcal{U}$ w.r.t. $K$ | (k) $\mathcal{U}$ w.r.t. $d$ | (l) $\mathcal{U}$ w.r.t. $N$ |

| (m) $\mathcal{C}$ w.r.t. $K$ | (n) $\mathcal{C}$ w.r.t. $d$ | (o) $\mathcal{C}$ w.r.t. $N$ | (p) $\mathcal{C}$ w.r.t. $K$ | (q) $\mathcal{C}$ w.r.t. $d$ | (r) $\mathcal{C}$ w.r.t. $N$ |

Figure 10: Quantitative analyses of the criterion triple when $\mathcal{P}_A$ and $\mathcal{P}_B$ are Gaussian distributions.

## F Supplementary Quantitative Analyses on Distribution Matching: Further Supporting the Main Findings in Section 2.3

To further elucidate the effects of the distributional matching, we conduct more quantitative analyses centered around the criterion triple $(\mathcal{E}, \mathcal{U}, \mathcal{C})$.

### F.1 Codebook Distribution and Feature Distribution are Gaussian Distributions

We begin by assuming that the distributions $\mathcal{P}_A$ and $\mathcal{P}_B$ are Gaussian. We generate a set of feature vectors $\{z_i\}_{i=1}^N$ from $\mathcal{N}_d(\mathbf{0}, \boldsymbol{I})$ and a set of code vectors $\{e_k\}_{k=1}^K$ from $\mathcal{N}_d(\mu \cdot \mathbf{1}, \boldsymbol{I})^3$, with $\mu$ varying within $\{0.0, 0.5, 1.0, 1.5, 2.0, 2.5\}$. The criterion triple results are presented in Figures 10(a) to 10(c), Figures 10(g) to 10(i), and Figures 10(m) to 10(o). Across all tested configurations of $K, d, N$, we consistently observe that when $\mu = 0$ — indicating identical distributions between $\mathcal{P}_A$ and $\mathcal{P}_B$ — the criterion triple achieves the lowest $\mathcal{E}$, highest $\mathcal{U}$, and largest $\mathcal{C}$. This empirical evidence reinforces the effectiveness of aligning feature and codebook distributions in VQ.

Additionally, we further analyze the criterion triple by varying the covariance matrix. We sample a set of feature vectors $\{z_i\}_{i=1}^N$ from the distribution $\mathcal{N}_d(\mathbf{0}, \boldsymbol{I})$ and a set of code vectors $\{e_k\}_{k=1}^K$ from $\mathcal{N}_d(\mathbf{0}, \sigma^2 \boldsymbol{I})$, where $\sigma$ is selected from $\{1, 2, 3, 4, 5, 6\}$. The results for the criterion triple are shown in Figures 10(d) to 10(f), Figures 10(j) to 10(l), and Figures 10(p) to 10(r). When $\sigma = 1$, indicating identical distributions between $\mathcal{P}_A$ and $\mathcal{P}_B$, all three evaluation criteria reach their optimal values: the lowest $\mathcal{E}$, highest $\mathcal{U}$, and largest $\mathcal{C}$ across all tested values of $K, d, N$. This result corroborates our earlier findings.

### F.2 Codebook Distribution and Feature Distribution are Unifrom Distributions

The above conclusion holds when $\mathcal{P}_A$ and $\mathcal{P}_B$ are other types of distributions, such as the uniform distribution. As shown in Figure 11, we sample a set of feature vectors $\{z_i\}_{i=1}^N$ from the distribution $\mathrm{Unif}_d(-1, 1)$ and a set of code vectors $\{e_k\}_{k=1}^K$ from $\mathrm{Unif}_d(\nu - 1, \nu + 1)$, where $\nu$ is selected from the set $\{0.0, 0.5, 1.0, 1.5, 2.0, 2.5\}$ or from $\mathrm{Unif}_d(-\zeta, \zeta)$, with $\zeta$ drawn from the set $\{1, 2, 3, 4, 5, 6\}$. We observe that when $\mu = 0$ or $\zeta = 1$—indicating that $\mathcal{P}_A$ and $\mathcal{P}_B$ have identical distributions—the performance in terms of the criterion triple is optimal, achieving the lowerest $\mathcal{E}$, the highest $\mathcal{U}$, and the largest $\mathcal{C}$ across all tested values of $K, d, N$. Therefore, we conclude that our quantitative analyses are distribution-agnostic and can be generalized to other distributions.

---

[3]$\mathbf{1}$ represents the vector of all ones.

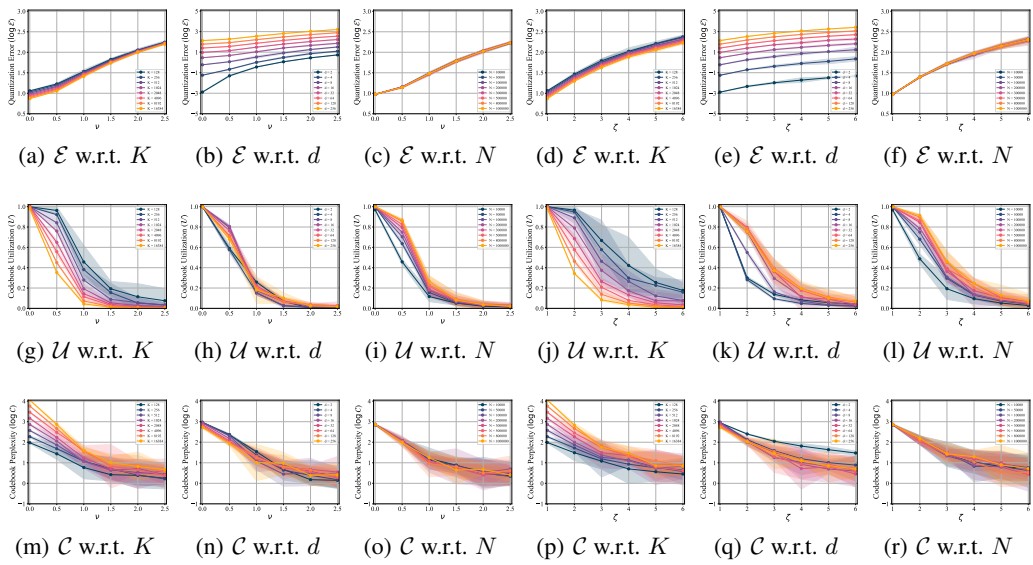

| (a) $\mathcal{E}$ w.r.t. $K$ | (b) $\mathcal{E}$ w.r.t. $d$ | (c) $\mathcal{E}$ w.r.t. $N$ | (d) $\mathcal{E}$ w.r.t. $K$ | (e) $\mathcal{E}$ w.r.t. $d$ | (f) $\mathcal{E}$ w.r.t. $N$ |

| (g) $\mathcal{U}$ w.r.t. $K$ | (h) $\mathcal{U}$ w.r.t. $d$ | (i) $\mathcal{U}$ w.r.t. $N$ | (j) $\mathcal{U}$ w.r.t. $K$ | (k) $\mathcal{U}$ w.r.t. $d$ | (l) $\mathcal{U}$ w.r.t. $N$ |

| (m) $\mathcal{C}$ w.r.t. $K$ | (n) $\mathcal{C}$ w.r.t. $d$ | (o) $\mathcal{C}$ w.r.t. $N$ | (p) $\mathcal{C}$ w.r.t. $K$ | (q) $\mathcal{C}$ w.r.t. $d$ | (r) $\mathcal{C}$ w.r.t. $N$ |

Figure 11: Quantitative analyses of the criterion triple when $\mathcal{P}_A$ and $\mathcal{P}_B$ are uniform distributions.

## G  The Significant Impact of Distribution Variance on Quantization Error

As discussed in Section 2.3 and 2.4, the optimal criterion triple is achieved when the distributions $\mathcal{P}_A$ and $\mathcal{P}_B$ are identical. In this section, we further analyze the criterion triple by the lens of distribution variance under the condition that both distributions are identical. Specifically, we first sample a set of feature vectors $\{z_i\}_{i=1}^N$ along with a set of code vectors $\{e_k\}_{k=1}^K$ from the distribution $\mathcal{N}_d(\mathbf{0}, \sigma^2 \mathbf{I})$ or the distribution $\mathrm{Unif}_d(-\zeta, \zeta)$. We then calculate the evaluation criteria according to their definitions in Section 2.2. As demonstrated in Table 4, $\sigma$ and $\zeta$ have a substantial impact on $\mathcal{E}$, while $\mathcal{U}$ and $\mathcal{C}$ remains largely unaffected.

Table 4: The criterion triple influence by the distribution variance.

| Evaluation Criteria | $\sigma$ | | | | | $\zeta$ | | | | |
|---|---|---|---|---|---|---|---|---|---|---|
| | 0.0001 | 0.001 | 0.01 | 0.1 | 1.0 | 0.0001 | 0.001 | 0.01 | 0.1 | 1.0 |
| $\mathcal{E}$ | 1.25e-8 | 1.25e-6 | 1.25e-4 | 1.24e-2 | 1.25 | 3.27e-9 | 3.27e-7 | 3.27e-5 | 3.27e-3 | 0.327 |
| $\mathcal{U}$ | 0.9934 | 0.9938 | 0.9940 | 0.9934 | 0.9941 | 0.9993 | 0.9986 | 0.9990 | 0.9992 | 0.9989 |
| $\mathcal{C}$ | 7265.3 | 7260.3 | 7267.7 | 7255.0 | 7275.8 | 7380.2 | 7372.2 | 7387.9 | 7397.5 | 7391.6 |

This experimental finding suggests that when the distribution variance of the feature vectors is uncontrollable or unknown, reporting a comparison of quantization error among various VQ methods is unreasonable. This is because the improvement in quantization error is predominantly attributed to the reduction in distribution variance rather than the effectiveness of the VQ methods. To evaluate various VQ methods in terms of the criterion triple, we establish an atomic and fair experimental setting in Appendix H, where the feature distributions for all VQ methods are identical.

## H  A Fair Setting to Evaluate Criterion Triple Evaluation

The distribution variance has a substantial impact on $\mathcal{E}$, as detailed in Appendix G. Therefore, the comparison of the quantization error among various VQ methods is unreasonable when the variance of the feature vectors is uncontrollable or unknown. This is because any improvement in quantization error is primarily attributed to the variance reduction rather than the inherent effectiveness of the VQ methods. To ensure a fair criterion triple evaluation, we provide a controlled experimental setting.

Specifically, we fix the feature distributions for all VQ methods to the same Gaussian distributions by setting $z_i \sim \mathcal{N}_d(\mu \cdot \mathbf{1}, \mathbf{I})$. Additionally, we initialize the codebook distribution as the standard Gaussian distribution across all VQ methods by sampling $e_k \sim \mathcal{N}_d(\mathbf{0}, \mathbf{I})$. In this experimental setup, the distribution variance is controlled to be the same for all VQ methods.

Our baseline includes Vanilla VQ [36], EMA VQ [29], Online VQ [42], and Linear VQ (a linear layer projection for frozen code vectors) [43, 44]. In all VQ algorithms, we treat the sampled code vectors

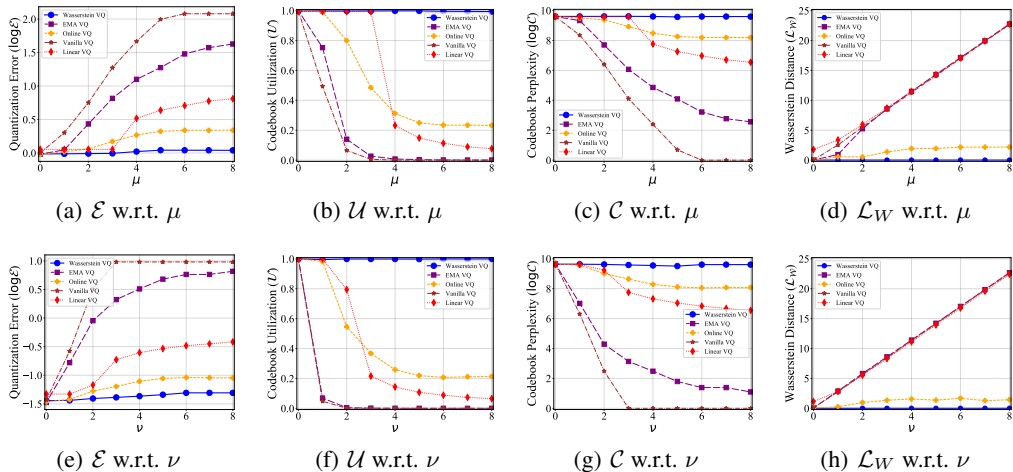

| (a) $\mathcal{E}$ w.r.t. $\mu$ | (b) $\mathcal{U}$ w.r.t. $\mu$ | (c) $\mathcal{C}$ w.r.t. $\mu$ | (d) $\mathcal{L}_W$ w.r.t. $\mu$ |
|---|---|---|---|
| (e) $\mathcal{E}$ w.r.t. $\nu$ | (f) $\mathcal{U}$ w.r.t. $\nu$ | (g) $\mathcal{C}$ w.r.t. $\nu$ | (h) $\mathcal{L}_W$ w.r.t. $\nu$ |

Figure 12: The performance metrics $(\mathcal{E}, \mathcal{U}, \mathcal{C})$ for various VQ approaches. For panels (a) to (d), the codebook distribution is initialized as a Gaussian distribution, while for panels (e) to (h), the codebook distribution is initialized as a uniform distribution.

as trainable parameters and optimize them using the respective algorithms. See detailed experimental specifications in Appendix I.4.

As visualized in Figures 12(a) to 12(c), *Wasserstein VQ* outperforms all baselines in terms of the criterion triple $(\mathcal{E}, \mathcal{U}, \mathcal{C})$, especially when the feature distribution and the initialized codebook distribution have large deviations. Although existing VQ methods can perform well with $\mu = 0$, this scenario is impractical, as the feature distribution is unknown and changes dynamically during training. When there is a large initial distribution gap between the codebook and the features (at $\mu = 5$), all existing VQ methods perform poorly. This indicates that the effectiveness of existing VQ methods is heavily dependent on their codebook initialization.

As observed in Figure 12(d), the Wasserstein distance of all existing VQ methods is obviously larger compared to that of *Wasserstein VQ* when $\mu \geq 1.0$, indicating that existing methods cannot achieve effective distribution alignment between features and the codebook. Conversely, *Wasserstein VQ* eliminates the reliance on codebook initialization via proposed explicit distributional matching regularization, thereby delivering the best performance in criterion metrics.

We can arrive at the same conclusion when the feature and codebook distributions are uniform, in which feature vectors are generated from $\text{Unif}_d(\nu - 1, \nu + 1)$ and code vectors are initialized from $\text{Unif}_d(-1, 1)$. As shown in Figures 12(e) to 12(h), *Wasserstein VQ* performs the best. This suggests that, despite being based on Gaussian assumptions, the effectiveness of our method exhibits a certain degree of distribution-agnostic behavior.

# I The Details of Synthetic Experiments

## I.1 Experimental Details in Section 2.3

As depicted in Figure 3 in Section 2.3, we conduct a qualitative analyses of the criterion triple. Specifically, we sample a set of feature vectors $\{z_i\}_{i=1}^N$ from within the red circle, and a collection of code vectors $\{e_k\}_{k=1}^K$ from within the green circle, with parameters set to $K = 400$, $N = 10000$ and $d = 2$ for the calculation of the criterion triple $(\mathcal{E}, \mathcal{U}, \mathcal{C})$. For the visualization, we select 10% of the feature vectors and 90% of the code vectors for plotting.

## I.2 Experimental Details in Appendix F

As illustrate in Figure 10 in Appendix F.1, we undertake comprehensive quantitative analyses centered around the criterion triple $(\mathcal{E}, \mathcal{U}, \mathcal{C})$. In these analyses, we assume that $\mathcal{P}_A$ and $\mathcal{P}_B$ are Gaussian distributions, from which we sample a set of feature vectors $\{z_i\}_{i=1}^N$ and a collection of code vectors

Table 5: Hyperparameters for the experiments in Table 1, 2, 3, and Table 6 and 7.

| Frameworks | VQ-VAE | VQ-VAE | VQ-VAE | VQGAN |
|---|---|---|---|---|
| Dataset | CIFAR-10/SVHN | FFHQ | ImageNet | FFHQ |
| Input size | $32 \times 32 \times 3$ | $256 \times 256 \times 3$ | $256 \times 256 \times 3$ | $256 \times 256 \times 3$ |
| Latent size | $8 \times 8 \times 8$ | $16 \times 16 \times 8$ | $16 \times 16 \times 8$ | $16 \times 16 \times 8$ |
| encoder/decoder channels | 64 | 64 | 64 | 128 |
| encoder/decoder channel mult. | $[1, 1, 2]$ | $[1, 1, 2, 2, 4]$ | $[1, 1, 2, 2, 4]$ | $[1, 1, 2, 2, 4]$ |
| Batch size | 128 | 32 | 32 | 64 |
| Initial Learning rate $lr$ | $5 \times 10^{-5}$ | $5 \times 10^{-5}$ | $5 \times 10^{-5}$ | $5 \times 10^{-4}$ |
| Codebook Loss Coefficient | 1.0 | 1.0 | 1.0 | 1.0 |
| Perceptual loss Coefficient | 0 | 0 | 0.0 | 1.0 |
| Adversarial loss Coefficient | 0 | 0 | 0.0 | 0.2 |
| Codebook dimensions | 8 | 8 | 8 | 8 |
| Training Epochs | 50 | 30 | 4 | 200 |
| GPU Resources | V100 16GB | A100 40GB | A100 40GB | 2 A100 80GB |

$\{e_k\}_{k=1}^K$. The default parameters are set to $N = 200,000$, $K = 1024$, and $d = 32$ for all figures unless otherwise specified. For instance, in Figure 10(a), $N$ and $d$ are taken at their default values, while the $K$ is varied within the set $\{128, 256, 512, 1024, 2048, 4096, 8192, 16284\}$. Additionally, each synthetic experiment is repeated five times, and the average results are reported, along with the calculation of 95% confidence intervals. In all figures, mean results are represented by points, while the confidence intervals are shown as shaded areas. Identical parameter settings are employed when $\mathcal{P}_A$ and $\mathcal{P}_B$ are uniform distributions, as illustrated in Figure 11 in Appendix F.2.

### I.3 Experimental Details in Appendix G

We set $K = 8192, d = 8, N = 100000$ when calculating the criterion triple $(\mathcal{E}, \mathcal{U}, \mathcal{C})$ in Appendix G. Each synthetic experiment is repeated five times, and the average results are reported in Table 4.

### I.4 Experimental Details in Appendix H

We provide experimental details of Figure 12 in Appendix H. In our experimental setup, we evaluate five distinct VQ algorithms using the criterion triple $(\mathcal{E}, \mathcal{U}, \mathcal{C})$. All experiments run on a single NVIDIA A100 GPU, with a codebook size $K$ of 16,384 and dimensionality $d$ of 8 across all algorithms. Each algorithm trains for 2,000 steps, with 50,000 feature vectors sampled from the specified Gaussian distribution at each step. For *Wasserstein VQ*, Vanilla VQ, and VQ + MLP, we use the SGD optimizer for training. For VQ EMA and Online Clustering, we use classical clustering algorithms—$k$-means [5] and $k$-means++[1]—to update code vectors.

## J  Experimental Details in Section 4

**Data Augmentation**  For FFHQ and ImageNet-1k datasets, we follow LLama Gen [33] and apply iterative box downsampling to resize images to 256×256 resolution. For CIFAR-10 and SVHN, the images are kept at their original resolution. Details are provided in Table 5.

**Encoder-Decoder Architecture**  For the ImageNet and FFHQ datasets, within both the VQ-VAE and VQGAN frameworks, our proposed Wasserstein VQ and all baseline methods adopt identical encoder-decoder architectures and parameter configurations, following the original VQGAN implementation [9]. Across all baselines in these frameworks, the encoder—a U-Net [31]—downscales the input image by a factor of 16. For CIFAR-10 and SVHN datasets, the encoder reduces the input resolution by a factor of 4. Further details are provided in Table 5.

**Training Details**  All experiments employ identical training settings: we use the AdamW optimizer [23] with $\beta_1 = 0.9$ and $\beta_1 = 0.95$, an initial learning rate $lr$, and apply a half-cycle cosine decay schedule following a linear warm-up phase. For specific details on training epochs and batch sizes, refer to Table 5.

**Loss Weight**  For all three baselines, $\beta$ is typically set to a value within the range $[0.25, 2]$. In our experiments, $\beta$ is set to a fixed value of $1.0$. For our proposed *Wasserstein VQ* model, we set

Table 6: Comparison of VQ-VAEs trained on CIFAR-10 dataset following [36].

| Approach | Tokens | Codebook Size | $\mathcal{U}$ (↑) | $\mathcal{C}$ (↑) | PSNR(↑) | SSIM(↑) | Rec. Loss (↓) |
|---|---|---|---|---|---|---|---|
| Vanilla VQ | 64 | 8192 | 2.7% | 186.9 | 27.15 | 0.83 | 0.0147 |
| EMA VQ | 64 | 8192 | 99.7% | 6416.1 | 29.43 | 0.88 | 0.0095 |
| Online VQ | 64 | 8192 | 22.1% | 995.4 | 28.20 | 0.85 | 0.0123 |
| **Wasserstein VQ** | 64 | 8192 | **100.0%** | **7781.8** | **29.88** | **0.90** | **0.0085** |
| Vanilla VQ | 64 | 16384 | 1.6% | 220.3 | 27.36 | 0.84 | 0.0141 |
| EMA VQ | 64 | 16384 | 80.8% | 10557.3 | 29.43 | 0.88 | 0.0093 |
| Online VQ | 64 | 16384 | 13.4% | 798.5 | 27.54 | 0.82 | 0.0141 |
| **Wasserstein VQ** | 64 | 16384 | **100.0%** | **15583.7** | **30.19** | **0.90** | **0.0080** |
| Vanilla VQ | 64 | 32768 | 0.5% | 154.8 | 27.10 | 0.83 | 0.0150 |
| EMA VQ | 64 | 32768 | 54.4% | 14427.0 | 29.57 | 0.88 | 0.0091 |
| Online VQ | 64 | 32768 | 7.2% | 1556.0 | 28.84 | 0.87 | 0.0106 |
| **Wasserstein VQ** | 64 | 32768 | **99.0%** | **29845.1** | **30.63** | **0.91** | **0.0071** |

Table 7: Comparison of VQ-VAEs trained on SVHN dataset following [36].

| Approach | Tokens | Codebook Size | $\mathcal{U}$ (↑) | $\mathcal{C}$ (↑) | PSNR(↑) | SSIM(↑) | Rec. Loss (↓) |
|---|---|---|---|---|---|---|---|
| Vanilla VQ | 64 | 8192 | 8.1% | 533.1 | 37.81 | 0.97 | 0.0018 |
| EMA VQ | 64 | 8192 | 56.8% | 3363.0 | 40.38 | **0.98** | 0.0010 |
| Online VQ | 64 | 8192 | 27.8% | 1325.1 | 39.04 | 0.97 | 0.0016 |
| **Wasserstein VQ** | 64 | 8192 | **88.2%** | **6154.5** | **41.04** | **0.98** | **0.0009** |
| Vanilla VQ | 64 | 16384 | 3.4% | 446.0 | 37.87 | 0.97 | 0.0017 |
| EMA VQ | 64 | 16384 | 22.2% | 2593.8 | 40.19 | **0.98** | 0.0011 |
| Online VQ | 64 | 16384 | 13.5% | 1090.5 | 39.12 | 0.97 | 0.0014 |
| **Wasserstein VQ** | 64 | 16384 | **87.5%** | **11967.2** | **41.49** | **0.98** | **0.0008** |
| Vanilla VQ | 64 | 32768 | 1.8% | 467.5 | 37.87 | 0.97 | 0.0017 |
| EMA VQ | 64 | 32768 | 35.8% | 7662.9 | 40.25 | **0.98** | 0.0010 |
| Online VQ | 64 | 32768 | 7.0% | 1334.8 | 39.26 | 0.97 | 0.0014 |
| **Wasserstein VQ** | 64 | 32768 | **88.7%** | **24376.3** | **41.84** | **0.98** | **0.0008** |

$\beta$ to a much smaller value, e.g., $\beta = 0.1$ for VQ-VAE and VQGAN. The smaller $\beta$ values enable the Wasserstein distance to dominate the loss function, thereby more effectively narrowing the gap between the distributions.

# K   VQ-VAE Performance on CIFAR-10 and SVHN datasets

Due to space limitations in the main text, we have relocated the VQ-VAE evaluation on CIFAR-10 and SVHN datasets to the appendix. As demonstrated in Table 6 and 7, our Wasserstein VQ consistently outperforms all baselines across both datasets, achieving superior results on nearly all evaluation metrics regardless of codebook size. Notably, we observe that Wasserstein VQ fails to reach 100% codebook utilization on SVHN, which may be attributed to the dataset's limited diversity.

# L   Analyses on Codebook Size and Dimensionality

We investigate the impact of the codebook size $K$ on the performance of VQ by varying across a wide range: $K \in [1024, 2048, 4096, 8192, 16384, 50000, 100000]$. As shown in Table 1 and Table 8, the vanilla VQ model suffers from severe codebook collapse even with a relatively small $K$, such as $K = 1024$. In contrast, improved algorithms like EMA VQ and Online VQ can handle smaller codebook sizes effectively, but they still experience codebook collapse when $K$ is very large, e.g., $K \geq 50000$. Notably, the *Wasserstein VQ* model consistently maintains 100% codebook utilization, irrespective of the codebook size. This underscores the effectiveness of distributional matching via the quadratic Wasserstein distance in mitigating the issue of codebook collapse.

We further investigate the impact of codebook dimensionality $d$ on VQ performance. Conducting experiments on CIFAR-10 with dimensionality $d$ ranging from 2 to 32, our proposed Wasserstein VQ consistently outperforms all baselines regardless of dimensionality, as shown in Table 9. Notably, we observe the curse of dimensionality phenomenon—performance degrades as dimensionality increases. Vanilla VQ exhibits the most severe degradation, followed by EMA VQ and Online VQ, while our Wasserstein VQ shows only minimal codebook utilization reduction.

Table 8: Supplementary comparison of VQ-VAEs trained on FFHQ dataset following [36] w.r.t codebook size $K$.

| Approach | Tokens | Codebook Size | $\mathcal{U}$ (↑) | $\mathcal{C}$ (↑) | PSNR(↑) | SSIM(↑) | Rec. Loss (↓) |
|---|---|---|---|---|---|---|---|
| Vanilla VQ | 256 | 1024 | 51.7% | 446.2 | 27.64 | 73.0 | 0.0125 |
| EMA VQ | 256 | 1024 | 74.1% | 618.9 | 27.66 | 72.7 | 0.0125 |
| Online VQ | 256 | 1024 | **100.0%** | 759.3 | 28.08 | 74.0 | 0.0114 |
| *Wasserstein VQ* | 256 | 1024 | **100.0%** | **977.4** | **28.11** | **74.4** | **0.0112** |
| Vanilla VQ | 256 | 2048 | 27.6% | 453.0 | 27.78 | 73.8 | 0.0121 |
| EMA VQ | 256 | 2048 | **100%** | 1608.0 | **28.39** | 74.9 | **0.0107** |
| Online VQ | 256 | 2048 | **100%** | 1462.6 | 28.34 | 74.6 | 0.0108 |
| *Wasserstein VQ* | 256 | 2048 | **100%** | **1840.5** | 28.32 | **75.3** | **0.0107** |
| Vanilla VQ | 256 | 4096 | 12.5% | 435.0 | 27.84 | 73.7 | 0.0119 |
| EMA VQ | 256 | 4096 | 76.7% | 2443.1 | 28.49 | 75.0 | 0.0104 |
| Online VQ | 256 | 4096 | 70.7% | 1600.0 | 28.25 | 74.1 | 0.0110 |
| *Wasserstein VQ* | 256 | 4096 | **100%** | **3895.4** | **28.54** | **75.1** | **0.0102** |
| Vanilla VQ | 256 | 8192 | 5.6% | 398.1 | 27.69 | 73.5 | 0.0122 |
| EMA VQ | 256 | 8192 | 28.9% | 1839.2 | 28.39 | 74.8 | 0.0106 |
| Online VQ | 256 | 8192 | 34.9% | 1474.4 | 28.15 | 73.9 | 0.0113 |
| *Wasserstein VQ* | 256 | 8192 | **100%** | **7731.5** | **28.81** | **76.2** | **0.0099** |

Table 9: Analysis On codebook dimension by the comparison of VQ-VAEs trained on CIFAR-10 dataset following [36]. (The codebook size $K$ is fixed to 16384)

| Approach | Tokens | Codebook Dim | $\mathcal{U}$ (↑) | $\mathcal{C}$ (↑) | PSNR(↑) | SSIM(↑) | Rec. Loss (↓) |
|---|---|---|---|---|---|---|---|
| Vanilla VQ | 256 | 2 | 3.8% | 532.2 | 27.00 | 0.80 | 0.0162 |
| EMA VQ | 256 | 2 | 97.6% | **14460.3** | 27.25 | 0.80 | **0.0155** |
| Online VQ | 256 | 2 | 9.0% | 611.8 | 26.62 | 0.79 | 0.0178 |
| *Wasserstein VQ* | 256 | 2 | **99.3%** | 12278.9 | **27.30** | **0.81** | **0.0155** |
| Vanilla VQ | 256 | 4 | 1.3% | 176.7 | 27.15 | 0.83 | 0.0149 |
| EMA VQ | 256 | 4 | 99.8% | 13153.9 | 29.57 | 0.89 | 0.0092 |
| Online VQ | 256 | 4 | 11.1% | 877.7 | 26.69 | 0.79 | 0.0173 |
| *Wasserstein VQ* | 256 | 4 | **100.0%** | **15724.7** | **29.93** | **0.89** | **0.0087** |
| Vanilla VQ | 256 | 8 | 1.6% | 220.3 | 27.36 | 0.84 | 0.0141 |
| EMA VQ | 256 | 8 | 80.8% | 10557.3 | 29.43 | 0.88 | **0.0009** |
| Online VQ | 256 | 8 | 13.4% | 798.5 | 27.54 | 0.82 | 0.0141 |
| *Wasserstein VQ* | 256 | 8 | **100.0%** | **15583.7** | **30.19** | **0.90** | **0.0080** |
| Vanilla VQ | 256 | 16 | 1.1% | 150.8 | 27.05 | 0.83 | 0.0152 |
| EMA VQ | 256 | 16 | 32.5% | 4169.2 | 29.31 | 0.88 | 0.0099 |
| Online VQ | 256 | 16 | 18.2% | 2051.0 | 28.29 | 0.85 | 0.0122 |
| *Wasserstein VQ* | 256 | 16 | **99.2%** | **14832.2** | **30.27** | **0.91** | **0.0078** |
| Vanilla VQ | 256 | 32 | 0.7% | 94.37 | 26.67 | 0.81 | 0.0165 |
| EMA VQ | 256 | 32 | 7.0% | 942.7 | 28.24 | 0.85 | 0.0122 |
| Online VQ | 256 | 32 | 18.8% | 2278.0 | 28.92 | 0.87 | 0.0104 |
| *Wasserstein VQ* | 256 | 32 | **96.4%** | **14056.9** | **30.39** | **0.91** | **0.0076** |

## M  Discussion with VQ-WAE [37]

VQ-WAE [37] introduces an alternative approach to distributional matching by employing Optimal Transport to optimize codebook vectors. Compared with our proposed distributional matching method, there are three key differences.

**First, regarding theoretical contributions**: VQ-WAE [37] claims that achieving optimal transport (OT) between code vectors and feature vectors yields the best reconstruction performance. Their notion of optimality encompasses both the VQ process and the encoder-decoder reconstruction pipeline. While we contend that incorporating complex encoder-decoder functions renders rigorous theoretical analysis intractable, VQ-WAE nevertheless asserts this conclusion. In contrast, our work deliberately excludes encoder-decoder components, focusing solely on the VQ process, which admits rigorous mathematical modeling. Through our proposed criterion triple, we theoretically prove that distributional matching guarantees optimal performance.

**Second, regarding distribution modeling**: VQ-WAE [37] assumes both code vectors and feature vectors follow uniform discrete distributions, whereas our method models them as continuous distributions. Specifically, VQ-WAE [37] represents the distributions of feature vectors $\{z_i\}_{i=1}^N$ and

Table 10: Reconstruction performance (↓: the lower the better and ↑: the higher the better). †:Results cited from VQ-WAE [37]. Codebook size $K$ is fixed to 512.

| Dataset | Model | Tokens | SSIM ↑ | PSNR ↑ | LPIPS ↓ | Rec. Loss (↓) | Perplexity ↑ |
|---|---|---|---|---|---|---|---|
| CIFAR10 | VQ-VAE† | 64 | 70 | 23.14 | 0.35 | | 69.8 |
| | SQ-VAE† | 64 | 80 | 26.11 | 0.23 | | 434.8 |
| | VQ-WAE† | 64 | 80 | 25.93 | 0.23 | | 497.3 |
| | VQ-WAE (Our run) | 64 | 13 | 14.60 | 0.41 | 0.247 | 1.0 |
| | Vanilla VQ | 64 | 83 | 27.19 | 0.03 | 0.015 | 192.5 |
| | EMA VQ | 64 | 84 | 27.97 | 0.04 | 0.013 | 436.1 |
| | Online VQ | 64 | 84 | 27.87 | 0.04 | 0.013 | 451.4 |
| | Wasserstein VQ | 64 | 86 | 28.26 | 0.03 | 0.012 | 481.7 |
| SVHN | VQ-VAE† | 64 | 88 | 26.94 | 0.17 | | 114.6 |
| | SQ-VAE† | 64 | 96 | 35.37 | 0.06 | | 389.8 |
| | VQ-WAE† | 64 | 96 | 34.62 | 0.07 | | 485.1 |
| | VQ-WAE (Our run) | 64 | 25 | 15.87 | 0.26 | 0.2026 | 1.0 |
| | Vanilla VQ | 64 | 97 | 38.18 | 0.01 | 0.0016 | 407.1 |
| | EMA VQ | 64 | 97 | 38.35 | 0.01 | 0.0017 | 408.9 |
| | Online VQ | 64 | 97 | 38.54 | 0.01 | 0.0017 | 421.5 |
| | Wasserstein VQ | 64 | 97 | 38.25 | 0.01 | 0.0016 | 423.5 |

code vectors $\{e_k\}_{k=1}^K$ as empirical measures:

$$\mathcal{P}_A = \frac{1}{N}\sum_{i=1}^N \delta_{z_i}, \quad \mathcal{P}_B = \frac{1}{N}\sum_{k=1}^K \delta_{e_k} \tag{14}$$

where $\delta_{z_i}$ and $\delta_{e_k}$ denote Dirac delta functions centered at $z_i$ and $e_k$, respectively. To align $\mathcal{P}_A$ and $\mathcal{P}_B$, VQ-WAE formulates the OT problem as:

$$\min_{\mathbf{P}\in\Pi(\mathcal{P}_A,\mathcal{P}_B)} \sum_{i=1}^N \sum_{k=1}^K P_{ik}\|z_i - e_k\|^2,$$
$$\text{s.t.} \quad \mathbf{P}\mathbf{1}_K = \frac{1}{N}\mathbf{1}_N, \quad \mathbf{P}^\top\mathbf{1}_N = \frac{1}{K}\mathbf{1}_K, \quad P_{ik} \geq 0 \quad \forall i,k, \tag{15}$$

where $\mathbf{P}$ is the transport plan, and the feasible set is:

$$\Pi(\mathcal{P}_A,\mathcal{P}_B) = \left\{ \mathbf{P} \in \mathbb{R}_+^{N\times K} \,\middle|\, \mathbf{P}\mathbf{1}_K = \frac{1}{N}\mathbf{1}_N, \mathbf{P}^\top\mathbf{1}_N = \frac{1}{K}\mathbf{1}_K \right\} \tag{16}$$

In contrast, we simplify the distributional assumption by modeling $\mathcal{P}_A$ and $\mathcal{P}_B$ as Gaussian distributions.

**Third, regarding computational efficiency**, The OT problem in VQ-WAE is prohibitively complex, whereas our quadratic Wasserstein distance incurs minimal overhead. To mitigate complexity, VQ-WAE employs a Kantorovich potential network. However, upon reproducing their code (no official implementation was released; we derived it from their ICLR 2023 supplementary material[4]), we observed severe non-convergence—the method degenerated to using a single code vector, failing to achieve distributional matching. Notably, VQ-WAE underperformed all other VQ baselines (Table 10).

In comparison, our quadratic Wasserstein distance (Equation 4) requires only low-dimensional matrix operations (e.g., $d = 8$), achieving superior performance and effective matching (Figure 5).

---

[4]See `https://openreview.net/forum?id=Z8qk2iM5uLI`. We includes the reproduced code and training logs of VQ-WAE in our supplementary materials.

