# OpenReview forum: "Enhancing Vector Quantization with Distributional Matching: A Theoretical and Empirical Study"
_NeurIPS.cc/2025/Conference — Submitted to NeurIPS 2025_

### Official Review · Reviewer_aFfQ · 2025-06-23

**Clarity:** 3
**Significance:** 3
**Originality:** 3
**Rating:** 5
**Confidence:** 3

**Summary:**

This paper investigates the distribution mismatch between the VQ codebook and the encoder output features, which this paper identifies as a key cause of two well-known issues in VQ-VAE: training instability and codebook collapse. To address this, the authors introduce a regularization term based on the quadratic Wasserstein distance, encouraging alignment between the codebook distribution and the encoder feature distribution in the embedding space. This alignment aims to reduce quantization error while improving codebook utilization and perplexity. Empirical results show that the proposed method achieves 100% codebook utilization and outperforms baseline VQ approaches in terms of model performance.

**Questions:**

1. Some key insights are buried in the appendix and should be included in the main paper to strengthen the empirical justification. For example, Table 8 provides more useful information: When the codebook utilization rate is low, a common way is reducing the codebook size. So a proper comparison with other baseline VQ methods should be done when the codebook utilization rate is somehow large.

2. There may be a conflict between the reconstruction objective and the distribution alignment objective. The reconstruction objective wants to pull a speicific codebook vector ( $e_{j}$ ) close to an feature vector ( $z_{j}$ ), that it was assigned to. However, the distribution matching loss might want to push that same codebook vector ( $e_{j}$ ) to a completely different region of the latent space to fill a gap. This conflict may undermine convergence or stability and is not well addressed in the paper.

3. Does the proposed method preserve the semantic structure in the latent space, which is critical for generative tasks beyond reconstruction?

4. It is unclear whether the enforced alignment improves or harms model generalization. For instance, would a VQ-VAE or VQGAN model trained with this regularization generalize well to out-of-distribution datasets (e.g., from ImageNet to COCO or Objects365)?

5. What is the actual computational cost of computing the quadratic Wasserstein distance, and how does it compare to other alternatives in practice? This paper points out "For computational efficiency, we employ the quadratic Wasserstein distance ... other statistical distances, such as the Kullback-Leibler divergence ... computationally expensive" from line 190 to 195, but it is not clear what is the computational overhead or runtime performance of Wasserstein distance calculation.

**Ethical Concerns:**

["NO or VERY MINOR ethics concerns only"]

**Final Justification:**

After reviewing the authors’ rebuttal and considering the strength and weakness of this paper, I have revised my recommendation from borderline accept to accept.

Resolved issues: They clarify the synergy between reconstruction and distribution alignment objectives via theoretical and practical reasoning. Additionally, OOD transfer experiments demonstrate generalization benefits, and runtime benchmarks show manageable computational overhead of the Wasserstein distance compared to baselines.

**Limitations:**

Yes

**Quality:**

3

**Strengths And Weaknesses:**

**Strengths:**

1. Authors provide a well-motivated analysis of the distribution mismatch issue and its impact on reconstruction quality and training dynamics.

2. The proposed quadratic Wasserstein regularization is conceptually straightforward and easy to implement as an additional loss term.

3. The experiments cover multiple settings, including varying codebook sizes and latent dimensions, offering insight into the robustness and scalability of the approach (though many results are relegated to the appendix).

**Weaknesses:**

1. Important findings, such as the effect of codebook utilization and its relation to codebook size, are only discussed in the appendix.

2. There is no evaluation of how well the distribution alignment preserves semantic structure or affects generalization to new datasets.

3. The paper lacks analysis of the computational overhead associated with the chosen quadratic Wasserstein distance metric.

---

> ### Author Rebuttal · Authors · 2025-07-31
>
> **R 4.1 Some key insights are buried in the appendix and should be included in the main paper to strengthen the empirical justification. For example, Table 8 provides more useful information: When the codebook utilization rate is low, a common way is reducing the codebook size. So a proper comparison with other baseline VQ methods should be done when the codebook utilization rate is somehow large.**
>
> Thank you for the insightful suggestion and for pointing out the valuable observations in Appendix Table 8. We agree that the relationship between codebook size and utilization is important and will move the relevant content to the main paper to strengthen the empirical justification.
>
> **R 4.2 There may be a conflict between the reconstruction objective and the distribution alignment objective. The reconstruction objective wants to pull a speicific codebook vector close to an feature vector, that it was assigned to. However, the distribution matching loss might want to push that same codebook vector to a completely different region of the latent space to fill a gap. This conflict may undermine convergence or stability and is not well addressed in the paper.**
>
> We appreciate the reviewer raising the potential conflict between the reconstruction objective and the distribution alignment objective. This is an insightful theoretical consideration. However, our analysis shows that in practice applications, these objectives act synergistically rather than adversarially. Our justification is as follows:
> - **The Reality of Codebook Size vs. Feature Size**: In practical VQ applications, the codebook size is typically orders of magnitude smaller than the number of feature vectors encountered during training (e.g., millions of features vs. thousands of codes). The scenario where codebook size matches or exceeds feature size (where a specific code vector could ideally align perfectly with a single feature vector) is rarely applicable.
> - **Codebook Vectors as Prototypes**: Under the common constraint of limited codebook size, each codebook vector necessarily acts as a prototype, representing not a single feature vector but the center of a cluster of similar feature vectors.
> - **Optimizing Both Objectives**: Crucially, Sections 2.3 and 2.4 provide empirical evidence and theoretical analysis establishing that when the distribution of the codebook vectors aligns with the distribution of the features (the distribution alignment objective), the quantization error (the reconstruction objective) is minimized.
> - **Synergy, Not Conflict**: This finding demonstrates that the distribution alignment objective promotes the minimization of the reconstruction error. By encouraging the codebook to mirror the overall structure and density of the feature space, the distribution alignment loss ensures that code vectors are positioned optimally to effectively act as cluster centers. This minimizes the average distance from feature vectors to their assigned codes (reconstruction loss), even if it means an individual code might not reside exactly where a specific feature vector lies. Effectively, distribution alignment helps strategically place the prototypes to best cover the feature manifold and fill potential gaps, thereby reducing overall quantization error.
>
> **R 4.3 Does the proposed method preserve the semantic structure in the latent space, which is critical for generative tasks beyond reconstruction?**
>
> We posit that our Wasserstein VQ approach preserves the semantic structure as effectively as standard Vector Quantization (Vanilla VQ). This preservation occurs because:
>
> - **Architectural Neutrality**: The method imposes no additional constraints on the feature vectors themselves.
> - **Gradient Isolation**: Critically, the Wasserstein distance gradients update only the codebook vectors, not the feature vectors.
> - **Distribution Matching**: The core mechanism solely adjusts the code vectors to match the empirical distribution of the features, regardless of how those features evolve during training.
>
> Consequently, the semantic structure inherent in the feature space remains fundamentally unaltered by our Wasserstein distance objective.
>
> **Empirical Investigation**: We empirically examined this on CIFAR-10 using t-SNE visualizations (available at: https://anonymous.4open.science/r/neurips_wasserstein_vq-F358). The results show:
>
> - Neither Vanilla VQ, EMA VQ, Online VQ, nor our Wasserstein VQ exhibited clear semantic clustering.
> - This aligns with findings in recent literature (e.g., [1]), indicating that optimizing by the  visual reconstruction loss may fails to induce meaningful semantic structure—a challenge also exists in the continuous tokenizers.
> - Therefore, to achieve better semantic structuring, future work should explore self-supervised objectives as [1].
>
> **Conclusion**:
>
> Based on the t-SNE analysis and our gradient isolation mechanism, our method introduces no detrimental effects on semantic structure compared to baseline VQ approaches.
>
> [1] Representation Alignment for Generation: Training Diffusion Transformers Is Easier Than You Think, ICLR 2025.
>
> **R 4.4 It is unclear whether the enforced alignment improves or harms model generalization. For instance, would a VQ-VAE or VQGAN model trained with this regularization generalize well to out-of-distribution datasets (e.g., from ImageNet to COCO or Objects365)?**
>
> To thoroughly assess the generalization ability of our model with the enforced alignment, we conducted a transfer experiment. We utilized the pre-trained checkpoint from Table 2, which was obtained by training the model on the ImageNet dataset. Subsequently, we evaluated the model’s performance on the FFHQ dataset, which is out-of-distribution compared to ImageNet. The evaluation results are presented in the table below.
>
> | Approach | Tokens | Codebook Size | $\mathcal{U}$ | $\mathcal{C}$ | PSNR | SSIM | Rec.Loss |
> | --- | --- | --- | --- | --- | --- | --- | --- |
> | Vanilla VQ | 256 | 16384 | 1.7\% | 332.9 | 27.18 | 71.6 | 0.0138 |
> | EMA VQ | 256 | 16384 | 12.5\% | 1292.1 | 27.69 | 72.3 | 0.0126 |
> | Online VQ | 256 | 16384 | 33.2\% | 2794.9 | 27.91 | 72.7 | 0.0120 |
> | **Wasserstein VQ** | 256 | 16384 | **99.2\%** | **13975.4** | **28.62** | **75.0** | **0.0103** |
> |  |  |  |  |  |  |  |  |
>
> We compared these results with those in Table 1, where the model was both trained and evaluated on the FFHQ dataset. It was observed that, with the exception of Online VQ, the performance of the other three VQ methods showed a slight decline when transferred from ImageNet to FFHQ. However, even in the face of this distribution shift, Wasserstein VQ still demonstrated the best performance among all the methods in this transfer experiment.
>
> This indicates that the proposed distribution alignment objective does not harm the model’s generalization ability. In fact, Wasserstein VQ’s superior performance on the out-of-distribution FFHQ dataset, despite the minor decline in performance for some metrics, suggests that the enforced alignment may potentially contribute to better generalization. Overall, these results provide strong evidence that the model trained with our regularization can generalize well to out-of-distribution datasets.
>
> **R 4.5 What is the actual computational cost of computing the quadratic Wasserstein distance, and how does it compare to other alternatives in practice? it is not clear what is the computational overhead or runtime performance of Wasserstein distance calculation.**
>
> Thank you for raising this important concern. We acknowledge the importance of runtime efficiency and provide clarification below.
>
> **Runtime Comparison with Alternatives**:
>
> We set the feature dimension $d=8$ and the number of data samples $N=8192$. Our main focus is on the runtime of the forward and backward passes of the VQ module across three different codebook sizes. To ensure reliable and comprehensive data for a more accurate analysis of computational performance, each configuration was run 100 times on an A100 GPU.
>
> | Approaches | Dimension | Data Samples | Codebook Size | Times (second) |
> | --- | --- | --- | --- | --- |
> | Vanilla VQ | 8 | 8192 | 16384 | **0.184**|
> | EMA VQ | 8 | 8192 | 16384 | 0.265  |
> | Online VQ | 8 | 8192 | 16384 | 1.810 |
> | **Wasserstein VQ**| 8 | 8192 | 16384 | $\underline{0.294}$|
> | Vanilla VQ | 8 | 8192 | 50000 | **0.408**|
> | EMA VQ | 8 | 8192 | 50000 | 0.807 |
> | Online VQ | 8 | 8192 | 50000 | 5.438 |
> | **Wasserstein VQ** | 8 | 8192 | 50000 | $\underline{0.525}$ |
> | Vanilla VQ | 8 | 8192 | 100000 | **0.655**|
> | EMA VQ | 8 | 8192 | 100000 | 1.502 |
> | Online VQ | 8 | 8192 | 100000 | 10.64 |
> | **Wasserstein VQ** | 8 | 8192 | 100000 | $\underline{0.774}$ |
> |  |  |  |  |  |  |  |  |
>
> As shown in the table,  even at large codebook sizes, specifically when $K\geq 50000$, the running time of Wasserstein VQ is only marginally longer than that of Vanilla VQ. This indicates that the computational efficiency of Wasserstein VQ remains quite high, and the introduction of the quadratic Wasserstein distance does not result in a substantial time overhead. Notably, while Online VQ incurs significant runtime increases at large $𝐾$, Wasserstein VQ remains much more efficient, further highlighting its scalability.
>
> **Computational Explanation**:
>
> The computational overhead of calculating the quadratic Wasserstein distance is relatively low. This is because it only involves the inversion of an 8-dimensional matrix followed by several matrix multiplication operations**. All of them can be implemented using PyTorch’s built-in functions, such as `torch.linalg.eigh` and `torch.mm`. These optimized functions in PyTorch leverage the parallel computing capabilities of GPUs, further reducing the actual computational burden.
>
> In conclusion, both empirical runtime and theoretical analysis indicate that the quadratic Wasserstein distance is computationally efficient and practical for real-world applications.

---

> ### Comment · Reviewer_aFfQ · 2025-08-05
>
> I would like to express my appreciation for the significant effort you have dedicated to the rebuttal. The rebuttal has addressed my initial questions to a considerable extent. Based on the insights gained from your rebuttal, I would like to revise my initial rating. The additional clarity you have provided has enhanced my understanding of the strengths and contributions of your paper.

---

> > ### Author Response · Authors · 2025-08-07
> > **Grateful Feedback and Commitment to Manuscript Enhancement**
> >
> > Dear Reviewer aFfQ,
> >
> > Thank you so much for your great patience and your appreciation for our rebuttal! We are truly grateful for your thoughtful and constructive comments, which have significantly contributed to improving the clarity and depth of our work.
> >
> > Based on your valuable insights, I will incorporate a detailed discussion on semantic structure, generalization ability, and computational cost into the revised manuscript. These additions will further strengthen the quality of our paper and ensure a more comprehensive presentation of our contributions.
> >
> > We deeply appreciate your time and effort in reviewing our work. Thank you once again for your support and encouragement!
> >
> > Best regards,
> > Authors of Submission 13897

---

### Official Review · Reviewer_nFXi · 2025-07-02

**Clarity:** 3
**Significance:** 3
**Originality:** 3
**Rating:** 4
**Confidence:** 4

**Summary:**

The paper studies the vector quantization in visual generation models and addresses the two issues that quantization cells (codes) are not used uniformly. Some codes in the codebook might be seldomly used to encode vectors, or even are empty. The paper proposes to add a quadratic Wasserstein regularization to the VAE/GAN loss function to force the codebook to have high utilization rate and uniform allocation.

Experiments show that the proposed method improves the codebook utilization rate and also leads to better image reconstruction quality.

**Questions:**

See above

**Ethical Concerns:**

["NO or VERY MINOR ethics concerns only"]

**Quality:**

2

**Strengths And Weaknesses:**

S1. Good motivation and simple but effective idea.
S2. The experiments are solid and show good improvements.

W1. The method assume Gaussian distribution of the feature vectors. Although the authors provided some empirical justification on the datasets tested, this might not always be true in practice.
W2. I don't fully understand the theoretical analysis. What are the takeways of the theorems? What's the relationship with the proposed Wasserstein regularization?

---

> ### Author Rebuttal · Authors · 2025-07-31
>
> **R 3.1 The method assume Gaussian distribution of the feature vectors. Although the authors provided some empirical justification on the datasets tested, this might not always be true in practice.**
>
> Thank you for this important observation. We acknowledge that the Gaussian distribution assumption may not hold universally in practice. To address this limitation, we are exploring the use of Maximum Mean Discrepancy (MMD) as a more flexible alternative for distribution alignment. MMD operates within a Reproducing Kernel Hilbert Space (RKHS) and requires no explicit distributional assumptions.
>
> For feature vectors $X = \\{z_1, z_2, ..., z_N\\}$  sampled from $\mathcal{P}_A$ and code vectors $Y = \\{e_1, e_2, ..., e_K\\}$ from $\mathcal{P}_B$, the squared MMD is defined as:
> \begin{equation}
> \mathcal{MMD}^2(X, Y) = \frac{1}{N^2} \sum _{i=1}^N \sum _{j=1}^N k(z_i, z_j) + \frac{1}{K^2} \sum _{i=1}^K \sum _{j=1}^K k(e_i, e_j)  - \frac{2}{NK} \sum _{i=1}^N \sum _{j=1}^K k(z_i, e_j)
> \end{equation}
> Where $k$ is a kernel function. Crucially, $\mathcal{MMD}=0$ if and only if $\mathcal{P}_A=\mathcal{P}_B$, making it a powerful tool for distribution alignment without Gaussian constraints.
>
> While we focus on Wasserstein distance in the current work, we will include comprehensive experiments with MMD regularization in the revised manuscript to validate its effectiveness for non-Gaussian scenarios.
>
> **R 3.2 I don't fully understand the theoretical analysis. What are the takeways of the theorems? What's the relationship with the proposed Wasserstein regularization?**
>
> Thank you for your feedback. I’ll clarify the key takeaways from our theoretical analysis and explain their relationship with the proposed Wasserstein regularization.
>
> The main insight from our theoretical analysis is that the quantization error hits its minimum when the feature vectors and code vectors follow the same distribution. This fundamental understanding is the cornerstone of our subsequent theorems and regularization approach.
>
> **Key Takeaways from Theorems**
> - **Theorem 1**: It establishes that for the quantization error to be minimized, the supports of the code vector distribution and the feature vector distribution must be identical. In simpler terms, the set of values that the feature vectors can take and those that the code vectors can represent should match. This ensures that the code vectors cover the entire range of possible feature vector values, reducing the chances of quantization errors due to out-of-range values.
> - **Theorem 2**: This theorem further refines the relationship by proving that when the quantization error is at its lowest, there is a consistency between the density functions of the code vector distribution and the feature vector distribution. It means that not only should the value ranges match (as shown in Theorem 1), but the likelihood of different values within those ranges should also be similar for both distributions.
>
> **Relationship with Wasserstein Regularization**
>
> Our proposed Wasserstein regularization is directly inspired by these theoretical findings. The primary goal of this regularization technique is to align the distributions of the feature vectors and code vectors. By doing so, we can closely approximate the ideal scenario where the two distributions are the same, as described by our theoretical analysis.
>
> When the distributions of the feature vectors and code vectors are aligned, we can minimize the quantization loss introduced by the Vector Quantization (VQ) module. In essence, the Wasserstein regularization serves as a practical method to achieve the theoretical optimum, ensuring that the quantization process is as efficient as possible and that the information loss during quantization is minimized.
>
> In summary, our theoretical analysis provides the theoretical basis, and the Wasserstein regularization is the practical implementation to achieve the goal of minimizing quantization error.

---

> ### Author Response · Authors · 2025-08-08
> **Additional Clarification and Empirical Evidence in R 3.1 (page 1)**
>
> Dear Reviewer nFXi,
>
> Following our constructive discussion with Reviewer SyAK, we would like to provide additional clarification and empirical evidence to further address your concern in **R3.1** regarding the **Gaussian assumption** in our Wasserstein regularization.
>
> ### **Part I: Clarifying the Role of the Gaussian Assumption**
>
> We emphasize that while our method adopts a Gaussian assumption to derive a closed-form expression for the Wasserstein distance (WD), **this does not impose any constraint on the learned feature distribution**. This clarification is grounded in three key points:
>
> **First, we adopt a Gaussian approximation solely to derive a closed-form expression for the Wasserstein loss, not to impose a Gaussian assumption on the actual feature distribution.**
>
> The Gaussian assumption serves primarily to derive a computationally tractable form of the Wasserstein distance (WD). The general form for distributions $\mathbb{P}_r$ (features) and $\mathbb{P}_g$(codebook) is:
>
> \begin{equation} W_{p}(\mathbb{P}_r, \mathbb{P}_g) = \left( \inf _{\gamma \in \Pi(\mathbb{P}_r ,\mathbb{P}_g)} \mathbb{E} _{(x, y) \sim \gamma}\left[|x - y|^{p}\right]\right)^{1/p} \end{equation}
>
> Assuming Gaussian distributions $\mathcal{N}(\mu_1, \Sigma_1)$ and $\mathcal{N}(\mu_2, \Sigma_2)$ enables a closed-form simplification for p=2:
>
> \begin{equation} \sqrt{\Vert \mu_1 - \mu_2 \Vert^2_2 + \mathop{\mathrm{tr}}\left( {\Sigma_1}+ {\Sigma_2} - 2 (\Sigma_1^{\frac{1}{2}} {\Sigma_2} \Sigma_1^{\frac{1}{2}})^{\frac{1}{2}}\right)}. \end{equation}
>
> While this special form leverages the Gaussian identity to become a simple, differentiable loss with low computational overhead (cf. R 4.5), it acts only as a regularizer matching only the first- and second-order moments (means and covariances) between $\mathbb{P}_r$ and $\mathbb{P}_g$. Crucially, this objective regularizes the distributions based on their statistics; it does not force the underlying feature distribution $\mathbb{P}_r$ to be Gaussian. The latent feature space retains the flexibility to develop complex, potentially non-Gaussian structures beyond these moments.
>
> **Second, WD gradients only update codebook parameters, not feature representations.**
>
> In our implementation, we explicitly detach the feature distribution statistics $(\mu_1, \Sigma_1)$ before computing the WD loss. This ensures that gradients from the WD loss flow solely into updating the code vectors (i.e., the codebook distribution $\mathbb{P}_g$), aligning its mean and covariance moments with those of the current features. No gradient is backpropagated into the feature encoder via this loss. Therefore, while the codebook adapts to the feature distribution, the WD loss itself does not directly influence feature learning.
>
> **Third, Latent Features Exhibit Gaussian-Like Behavior Consistent with the Central Limit Theorem (CLT).**
>
> The approximately Gaussian distribution observed in the learned latent features appears to emerge naturally from the aggregation of many weakly dependent components in deep representations, a phenomenon broadly consistent with the CLT. As shown in Figure 5(a–c), even models trained without any WD regularization—such as Vanilla VQ, EMA VQ, and Online VQ—consistently produce Gaussian-like latent feature distributions. This suggests that the observed Gaussianity is a consequence of the inherent learning dynamics, rather than a result of the WD loss or its Gaussian assumption.
>
> **In summary,** the Gaussian assumption in the WD formulation serves a practical computational role and does not restrict the expressiveness or structure of the learned feature representations.
>
> **A final nuance:** We acknowledge that if the ideal feature distribution were highly non-Gaussian, matching only the first and second moments might be insufficient for accurate distributional alignment. In such rare cases (which our empirical results suggest are uncommon in practice), alternative objectives such as MMD could provide better fidelity. Nonetheless, given the approximate Gaussianity observed across various models, we find that the moment-matching WD loss remains an effective and lightweight mechanism for regularizing codebook distributions in VQ models.

---

> > ### Author Response · Authors · 2025-08-08
> > **Additional Clarification and Empirical Evidence in R 3.1 (page 2)**
> >
> > ### **Part II: Empirical Evaluation of the Gaussian Assumption**
> >
> > To further evaluate the assumption, we implemented **Maximum Mean Discrepancy (MMD)** regularization, which makes **no distributional assumptions**, and compared it with Wasserstein regularization across both **real-world** and **synthetic** datasets.
> >
> > **First, Real-World Datasets (FFHQ and ImageNet)**
> >
> > **On the FFHQ dataset:**
> > | Approach | Tokens | Codebook Size | $\mathcal{U}$ | $\mathcal{C}$ | PSNR | SSIM | Rec.Loss |
> > | --- | --- | --- | --- | --- | --- | --- | --- |
> > | Wasserstein VQ | 256 | 4096 | 100\% | 3895.4 | **28.54** | 75.1 | **0.0102** |
> > | MMD VQ | 256 | 4096 | 100\% | **3951.7** | **28.54** | **75.6** | 0.0103 |
> > | Wasserstein VQ | 256 | 8192 | 100\% | 7731.5 | **28.81** | **76.2** | 0.0099 |
> > | MMD VQ | 256 | 8192 | 100\% | **7933.3** | 28.76 | 76.0 | **0.0098** |
> > | Wasserstein VQ | 256 | 16384 | 100\% | **15713.3** | **29.03** | **76.6** | **0.0093** |
> > | MMD VQ | 256 | 16384 | 100\% | 15598.1 | 28.92 | 76.4 | 0.0096 |
> > |  |  |  |  |  |  |  |  |
> >
> > **On the ImageNet Dataset:**
> >
> > | Approach | Tokens | Codebook Size | $\mathcal{U}$ | $\mathcal{C}$ | PSNR | SSIM | Rec.Loss |
> > | --- | --- | --- | --- | --- | --- | --- | --- |
> > | Wasserstein VQ | 256 | 16384 | **100\%** | **15539.1** | 25.47 | 61.2 | 0.0242 |
> > | MMD VQ | 256 | 16384 | 99.8\% | 15370.2 | **25.49** | **61.7** | **0.0240** |
> > |  |  |  |  |  |  |  |  |
> >
> > **Key Observations:**
> >
> > - On FFHQ, MMD VQ achieves comparable or slightly improved SSIM but exhibits no clear advantage in PSNR/Rec.Loss.
> > - On ImageNet, MMD VQ shows marginal improvements in SSIM/rec. loss but reduced code utilization (99.8% vs. 100% for Wasserstein VQ).
> > - **Practical Limitation**: MMD VQ incurs prohibitive GPU memory costs for large codebooks (e.g., failed at K=50,000/100,000), whereas Wasserstein VQ scales better.
> >
> >  **Second, Controlled Simulation: Non-Gaussian vs. Gaussian Features**
> >
> > We also designed a synthetic experiment where features were explicitly sampled from uniform (non-Gaussian) or Gaussian distributions. Code vectors were optimized via five methods (d=8, K=16384):
> >
> > **Uniform Feature Distribution:**
> >
> > | Approach | $\mathcal{E}$ | $\mathcal{U}$ | $\mathcal{C}$ |
> > | --- | --- | --- | --- |
> > | Vanilla VQ | 2.67 | 0.01 \% | 1.00 |
> > | EMA VQ | 1.38 | 0.14 \% | 22.98 |
> > | Online VQ | 0.30 | 36.7 \% | 5670.4 |
> > | Wasserstein VQ | 0.25 | **99.9\%** | **14473.5** |
> > | MMD VQ | **0.24** | 98.7\% | 14454.3 |
> > |  |  |  |  |
> >
> > **Gaussian Feature Distribution:**
> >
> > | Approach | $\mathcal{E}$ | $\mathcal{U}$ | $\mathcal{C}$ |
> > | --- | --- | --- | --- |
> > | Vanilla VQ | 3.57 | 0.6\% | 60.96 |
> > | EMA VQ | 2.26 | 2.7\% | 433.5 |
> > | Online VQ | 1.19 | 48.4\% | 7499.6 |
> > | Wasserstein VQ | **0.99** | **99.8\%** | **14958.4** |
> > | MMD VQ | 1.01 | 99.5\% | 14680.5 |
> > |  |  |  |  |
> >
> > **Key Findings:**
> >
> > - In **non-Gaussian** (uniform) settings, MMD VQ slightly reduces quantization error (+0.01 improvement) but with marginally lower code utilization.
> > - In **Gaussian** settings, Wasserstein VQ outperforms MMD VQ in quantization error.
> > - Online VQ performed relatively well in this setting only due to static feature distributions; it is generally suboptimal in dynamic real-world data.
> >
> > ###
> >
> > ### **Conclusion**
> >
> > The empirical advantage of MMD VQ appears only under controlled, non-Gaussian settings, and the gain is minimal (approximately 4% in quantization error). For real-world datasets where features tend to follow approximately Gaussian distributions—likely due to the central limit effect in deep representations—Wasserstein VQ is preferable due to its consistent performance, higher code utilization, and significantly lower GPU memory demands. Thus, we find Wasserstein regularization to be a **practical and effective solution**, grounded in theory, and validated across diverse datasets.
> >
> > We sincerely welcome any further questions or concerns, and would be happy to address them before the discussion period concludes.
> >
> > Thank you again for your thoughtful and constructive feedback.
> >
> > Authors of Submission 13897

---

### Official Review · Reviewer_SyAK · 2025-07-02

**Clarity:** 3
**Significance:** 2
**Originality:** 2
**Rating:** 4
**Confidence:** 5

**Summary:**

The authors address two issues in training vector quantization (VQ) models: training instability and codebook collapse. They attribute these problems to a distributional mismatch between feature vectors and codebook entries. Assuming both follow Gaussian distributions, they propose regularizing the standard VQ-VAE/VQGAN training objectives with a quadratic Wasserstein distance. The method is evaluated empirically on standard image datasets and achieves near 100% codebook utilization with improved reconstruction quality compared to standard VQ training.

**Questions:**

- How does the proposed method help address training instability? The paper provides strong evidence of improved codebook utilization, but the connection to stabilizing training is less clear. Is the instability primarily a result of codebook collapse, where the model could trivially reduce quantization error by overusing a few entries?

- The choice of a Gaussian distribution for $P_z$ is understandable for simplicity, but as mentioned above, how sensitive is the method to deviations from this assumption? If the true feature distribution is significantly non-Gaussian, how well does the proposed feature matching loss still work, and what kind of degradation should we expect?

- In Table 2, with a 100k size codebook, Wasserstein VQ improves utilization by over 200× compared to Vanilla VQ, yet the gain in reconstruction quality appears marginal. It seems to me the vanilla VQ, which has no explicit distributional assumption, could still achieve good reconstruction even with much fewer codebook vectors. Could the Gaussian constraint on $z_e$ be reducing its expressiveness, thereby capping the performance?

- Since distribution matching is the major contribution, would the authors consider extending the Gaussian assumption to a more expressive model, such as a Gaussian mixture? If this extension leads to stronger performance, it would further strengthen the soundness of the proposed method.

- [1] also assumes a Gaussian distribution over the codebook but employs an affine reparameterization of the codebook vectors to improve gradient flow. How does the proposed distribution matching compare to this alternative?

[1] Straightening out the straight-through estimator: Overcoming optimization challenges in vector quantized networks. ICML, 2023.

**Ethical Concerns:**

["NO or VERY MINOR ethics concerns only"]

**Final Justification:**

The authors have addressed most of my initial doubts. While the performance improvement is still not particularly large, the paper offers a novel perspective on training VQ-VAE models. The additional results provided in the rebuttal place the paper within the acceptance range.

**Limitations:**

The authors have discussed limitations of their study.

**Quality:**

3

**Strengths And Weaknesses:**

Strengths:
- The use of the Wasserstein distance to align feature and codebook distributions is interesting and theoretically supported.
- The approach is straightforward to implement under the Gaussian assumption.
- The empirical results are comprehensive and demonstrate clear improvements in codebook utilization and, to some extent, reconstruction quality.

Weaknesses:
- The improvement in reconstruction quality appears marginal, especially given the substantial increase in codebook utilization.
- The method depends heavily on the Gaussian assumption for the feature distribution, which may not hold in general. The paper does not explain or tackle how the method performs when the assumption is violated or when the feature distribution is more complex.

---

> ### Author Rebuttal · Authors · 2025-07-31
>
> **R 2.1 How does the proposed method help address training instability? The paper provides strong evidence of improved codebook utilization, but the connection to stabilizing training is less clear. Is the instability primarily a result of codebook collapse, where the model could trivially reduce quantization error by overusing a few entries?**
>
> **First: Minimizing quantization error is key to addressing training instability.**
> As discussed in lines 83–89, training instability primarily stems from the *gradient gap* introduced by the Straight-Through Estimator (STE) in the VQ module. In STE, gradients are directly copied from $z_q$ to $z_e$, but when the two are far apart, this approximation becomes inaccurate and leads to unstable training. To mitigate this, we aim to minimize the distance between $z_q$ and $z_e$, thereby reducing the gradient mismatch and improving training stability.
>
> **Second: Minimizing quantization error necessarily requires 100% codebook utilization (i.e., no collapse).**
> We would like to clarify that minimizing quantization error necessarily results in full codebook utilization. In other words, if codebook collapse exists, it is impossible to achieve minimal quantization error. However, the reverse does not hold: 100% codebook utilization does not guarantee minimal quantization error or training stability.
>
> To illustrate this, we provide a simple proof by contradiction. Suppose there are $n$ code vectors, but only $n-1$ are being used. This implies that one code vector, say $v_k$, is never the nearest to any feature vector. However, we can always construct a new code vector $v_k'$ that is closer to at least one feature vector than the currently assigned one, thereby reducing the overall quantization error. Consequently, $v_k'$ would be used, contradicting the assumption that only $n-1$ code vectors are active. Hence, minimizing quantization error implies that all $n$ code vectors must be used.
>
> If needed, we are happy to provide a formal proof during the discussion phase. In short, minimizing information loss naturally leads to full codebook utilization, but full utilization alone does not guarantee low quantization error or stable training.
>
> **Third: Aligning the distributions of $z_q$ and $z_e$ minimizes their distance.**
> As shown in Section 2 through both theoretical analysis and empirical results, aligning the distributions of $z_q$ and $z_e$ minimizes their distance. Our proposed Wasserstein Distance facilitates this alignment, thereby reducing the gap between $z_q$ and $z_e$ and enhancing training stability.
>
> **Fourth: Empirical validation under controlled distribution variance.**
> As shown in Appendix G, we found that the quantization error (i.e., $\|z_q - z_e\|$) is highly sensitive to the distribution variance of $z_e$, which varies across different VQ methods. To ensure a fair comparison, we control this variance in our experiments in Appendix H and show that the proposed Wasserstein distance consistently achieves the lowest quantization error among all baselines, further validating its effectiveness.
>
> **In summary**, training instability stems primarily from the gradient gap induced by quantization error, not codebook collapse alone. However, minimizing this error inherently prevents collapse. Our proposed Wasserstein Distance reduces quantization error both theoretically and empirically, thereby enhancing training stability and model performance in a principled and generalizable way.
>
> **R 2.2 The choice of a Gaussian distribution for is understandable for simplicity, but as mentioned above, how sensitive is the method to deviations from this assumption? If the true feature distribution is significantly non-Gaussian, how well does the proposed feature matching loss still work, and what kind of degradation should we expect?**
>
> Thank you for the thoughtful question. We acknowledge that analyzing sensitivity to deviations from the Gaussian assumption is important but nontrivial. Currently, we have not established an empirical framework to evaluate this systematically. If a feasible experimental setup is suggested, we would be happy to explore it during the discussion phase.
>
> In practice, we observe that the feature vectors in our VQ-VAE experiments are approximately Gaussian (see Figure 5 and 6), indicating that the assumption is reasonable in our current settings. Exploring more expressive priors is an interesting direction for future work.
>
> **R 2.3 In Table 2, ...,Wasserstein VQ improves utilization ..., yet the gain ... appears marginal. It seems to me the vanilla VQ, which has no explicit distributional assumption, .... Could the Gaussian constraint on be reducing its expressiveness?**
>
> **Gaussian Constraint On Expressiveness:**
>
> Thank you for the insightful question. We believe the Gaussian constraint in our method does not reduce expressiveness:
>
> - **No Constraints on Features:** Unlike VAEs, we apply no reparameterization, and features are unconstrained, identical to Vanilla VQ.
> - **Code Vectors Only:**  Wasserstein gradients exclusively update code vectors, solely aligning the codebook’s distribution with features’. The representational capacity of the feature vectors themselves is unaffected.This design ensures the Gaussian approximation is solely a tool for efficient codebook learning, not a restrictive prior on the latent features.
>
> **Regarding Reconstruction Gain**:
>
> In our VQVAE experiment, we first tune Vanilla VQ to achieve a very good performance and obtain a set of hyperparameters. Then we **fix** these hyperparameters and add the Wasserstein distance as a constraint to Vanilla VQ to get our experimental results. This fixed setup may limit observed gains; we expect further improvements with joint tuning. Besides PSNR, our method shows consistent improvements across metrics—e.g., ~+4 dB SSIM, 100% codebook utilization, and richer visual details—on multiple datasets. It is also lightweight, plug-and-play, and integrates easily into existing frameworks, making it broadly practical and effective.
>
>
>
> **R 2.4 Since distribution matching is the major contribution, would the authors consider extending the Gaussian assumption to a more expressive model, such as a Gaussian mixture? If this extension leads to stronger performance, it would further strengthen the soundness of the proposed method.**
>
> Thank you for the insightful suggestion. To this end, we are exploring Maximum Mean Discrepancy (MMD) as a more general alternative for distribution alignment. MMD operates within a Reproducing Kernel Hilbert Space (RKHS) and requires no explicit distributional assumptions.
>
> For feature vectors $X = \\{z_1, z_2, ..., z_N\\}$  sampled from $\mathcal{P}_A$ and code vectors $Y = \\{e_1, e_2, ..., e_K\\}$ from $\mathcal{P}_B$, the squared MMD is:
> \begin{equation}
> \mathcal{MMD}^2(X, Y) = \frac{1}{N^2} \sum _{i=1}^N \sum _{j=1}^N k(z_i, z_j) + \frac{1}{K^2} \sum _{i=1}^K \sum _{j=1}^K k(e_i, e_j)  - \frac{2}{NK} \sum _{i=1}^N \sum _{j=1}^K k(z_i, e_j)
> \end{equation}
> Where $k$ is a kernel function. Crucially, $\mathcal{MMD}=0$ if and only if $\mathcal{P}_A=\mathcal{P}_B$, making it a powerful tool for distribution alignment without Gaussian constraints.  We will include comprehensive experiments with MMD regularization in the revised manuscript to validate its effectiveness for non-Gaussian scenarios. We have not yet found a feasible way to incorporate a Gaussian mixture into the VQ module, but we would be glad to explore any suggestions.
>
> **R 2.5 [1] also assumes a Gaussian distribution over the codebook but employs an affine reparameterization of the codebook vectors. How does the proposed distribution matching compare to this alternative?**
>
> We add the STE++[1] experiments results on the FFHQ, ImageNet and CIFAR-10 dataset.
>
> FFHQ dataset:
>
> | Approach | Tokens | Codebook Size | $\mathcal{U}$ | $\mathcal{C}$ | PSNR | SSIM | Rec.Loss |
> | --- | --- | --- | --- | --- | --- | --- | --- |
> | STE++ | 256 | 16384 | 3.4\% | 476.7 | 27.54 | 72.3 | 0.0129 |
> | Wasserstein VQ | 256 | 16384 | 100\% | 15713.3 | 29.03 | 76.6 | 0.0093 |
> | STE++ | 256 | 50000 | 1.0\% | 447.2 | 27.49 | 72.4 | 0.0131 |
> | Wasserstein VQ | 256 | 50000 | 100\% | 47496.4 | 29.24 | 77.0 | 0.0089 |
> | STE++ | 256 | 100000 | 0.5\% | 450.7 | 27.52 | 72.4 | 0.0130 |
> | Wasserstein VQ | 256 | 100000 | 100\% | 93152.7 | 29.53 | 78.0 | 0.0083 |
> |  |  |  |  |  |  |  |  |
>
> ImageNet dataset:
>
> | Approach | Tokens | Codebook Size | $\mathcal{U}$ | $\mathcal{C}$ | PSNR | SSIM | Rec.Loss |
> | --- | --- | --- | --- | --- | --- | --- | --- |
> | STE++ | 256 | 16384 | 6.5\% | 889.9 | 24.88 | 58.9 | 0.0270 |
> | Wasserstein VQ | 256 | 16384 | 100\% | 15539.1 | 25.47 | 61.2 | 0.0242 |
> | STE++ | 256 | 50000 | 2.0\% | 851.7 | 24.89 | 59.0 | 0.0270 |
> | Wasserstein VQ | 256 | 50000 | 100\% | 46133.2 | 25.72 | 62.3 | 0.0230 |
> | STE++ | 256 | 100000 | 0.9\% | 730.6 | 24.86 | 59.1 | 0.0269 |
> | Wasserstein VQ | 256 | 100000 | 100\% | 93264.7 | 25.88 | 63.0 | 0.0223 |
> |  |  |  |  |  |  |  |  |
>
> We used the publicly available STE++ code, which provides only a partial implementation, and integrated it with our own LDM encoder-decoder structure using the default parameters. However, we were unable to fully reproduce the performance reported in [1]. In our experiments on ImageNet, Wasserstein VQ consistently outperforms STE++ across all metrics, including reconstruction quality and codebook utilization. To ensure transparency and reproducibility, we have released our code, scripts, logs, and result files on an anonymous GitHub repository: https://anonymous.4open.science/r/neurips_wasserstein_vq-F358.

---

> > ### Comment · Reviewer_SyAK · 2025-08-05
> >
> > Thank you for your response, which has helped address some of my concerns. However, I still have the following points:
> >
> > 1. Could you elaborate further on how the Gaussian assumption does not constrain the feature representation? From my understanding, the Wasserstein loss relies on the Gaussian assumption for both feature and code vectors, and it regularizes both distributions accordingly.
> >
> > 2. If fixed parameters are the reason for the limited reconstruction gain, could you provide results after tuning? From my perspective, the proposed Wasserstein regularization indeed effectively reduces quantization error and enables full codebook utilization. However, this benefit appears to rely on a simplified Gaussian assumption, which may limit the expressiveness of the model, potentially contributing to the marginal gains (compared to vanilla) reported in Table 2. It would make the paper’s contribution more complete if the authors could, for example, include a discussion of the method’s behavior under non-Gaussian features, provide results with a more expressive prior, or consider alternative distribution-matching techniques that do not rely on specific distributional assumptions (such as the suggested MMD regularization).
> >
> > 3. Regarding the "MMD regularization for non-Gaussian scenarios", could you summarize the relevant findings here as we won’t have access to your updated manuscript?

---

> ### Author Response · Authors · 2025-08-07
> **Further Discussion Page 1**
>
> **R 2.6 Could you elaborate further on how the Gaussian assumption does not constrain the feature representation? From my understanding, the Wasserstein loss relies on the Gaussian assumption for both feature and code vectors, and it regularizes both distributions accordingly.**
>
> We thank the reviewer for the insightful comment. We would like to clarify that while our method adopts a Gaussian assumption to derive a closed-form expression for the Wasserstein distance (WD), this does **not** functionally constrain the learned feature distribution. This clarification rests on three key points:
>
> **First, we adopt a Gaussian approximation solely to derive a closed-form expression for the Wasserstein loss, not to impose a Gaussian assumption on the actual feature distribution.**
>
> The Gaussian assumption serves primarily to derive a computationally tractable form of the Wasserstein distance (WD). The general form for distributions $\mathbb{P}_r$ (features) and $\mathbb{P}_g$(codebook) is:
>
> \begin{equation} W_{p}(\mathbb{P}_r, \mathbb{P}_g) = \left( \inf _{\gamma \in \Pi(\mathbb{P}_r ,\mathbb{P}_g)} \mathbb{E} _{(x, y) \sim \gamma}\left[|x - y|^{p}\right]\right)^{1/p} \end{equation}
>
> Assuming Gaussian distributions $\mathcal{N}(\mu_1, \Sigma_1)$ and $\mathcal{N}(\mu_2, \Sigma_2)$ enables a closed-form simplification for p=2:
>
> \begin{equation} \sqrt{\Vert \mu_1 - \mu_2 \Vert^2_2 + \mathop{\mathrm{tr}}\left( {\Sigma_1}+ {\Sigma_2} - 2 (\Sigma_1^{\frac{1}{2}} {\Sigma_2} \Sigma_1^{\frac{1}{2}})^{\frac{1}{2}}\right)}. \end{equation}
>
> While this special form leverages the Gaussian identity to become a simple, differentiable loss with low computational overhead (cf. **R 4.5**), **it acts only as a regularizer matching only the first- and second-order moments (means and covariances)** between $\mathbb{P}_r$ and $\mathbb{P}_g$**.** Crucially, this objective regularizes the distributions based on their statistics; it does not force the underlying feature distribution $\mathbb{P}_r$ to be Gaussian. The latent feature space retains the flexibility to develop complex, potentially non-Gaussian structures beyond these moments.
>
> **Second, WD Gradients Only Update Codebook Parameters, Not Feature Representations:**
>
> In our implementation, we explicitly detach the feature distribution statistics $(\mu_1, \Sigma_1)$ before computing the WD loss. This ensures that gradients from the WD loss flow solely into updating the code vectors (i.e., the codebook distribution $\mathbb{P}_g$), aligning its mean and covariance moments with those of the current features. **No gradient is backpropagated into the feature encoder via this loss.** Therefore, while the codebook adapts to the feature distribution, the WD loss itself does not directly influence feature learning.
>
> **Third, Latent Features Exhibit Gaussian-Like Behavior Consistent with the Central Limit Theorem (CLT):**
>
> The approximately Gaussian distribution observed in the learned latent features appears to emerge naturally from the aggregation of many weakly dependent components in deep representations, a phenomenon broadly consistent with the CLT. **As shown in Figure 5(a–c), even models trained without any WD regularization—such as Vanilla VQ, EMA VQ, and Online VQ—consistently produce Gaussian-like latent feature distributions. This suggests that the observed Gaussianity is a consequence of the inherent learning dynamics, rather than a result of the WD loss or its Gaussian assumption.**
>
> **In summary,** the Gaussian assumption in the WD formulation serves a practical computational role and does not restrict the expressiveness or structure of the learned feature representations.
>
> **A final nuance:** We acknowledge that if the ideal feature distribution were highly non-Gaussian, matching only the first and second moments might be insufficient for accurate distributional alignment. In such rare cases (which our empirical results suggest are uncommon in practice), alternative objectives such as MMD could provide better fidelity. Nonetheless, given the approximate Gaussianity observed across various models, we find that the moment-matching WD loss remains an effective and lightweight mechanism for regularizing codebook distributions in VQ models.

---

> ### Author Response · Authors · 2025-08-07
> **Further Discussion Page 2**
>
> **R 2.7 Regarding the "MMD regularization for non-Gaussian scenarios", could you summarize the relevant findings here as we won’t have access to your updated manuscript?**
>
> Thank you for your interest in our analysis of MMD regularization under non-Gaussian feature distributions. Below, we summarize key experimental findings on FFHQ and ImageNet datasets, along with controlled simulations:
>
> On the FFHQ dataset:
>
> | Approach | Tokens | Codebook Size | $\mathcal{U}$ | $\mathcal{C}$ | PSNR | SSIM | Rec.Loss |
> | --- | --- | --- | --- | --- | --- | --- | --- |
> | Wasserstein VQ | 256 | 4096 | 100\% | 3895.4 | **28.54** | 75.1 | **0.0102** |
> | MMD VQ | 256 | 4096 | 100\% | **3951.7** | **28.54** | **75.6** | 0.0103 |
> | Wasserstein VQ | 256 | 8192 | 100\% | 7731.5 | **28.81** | **76.2** | 0.0099 |
> | MMD VQ | 256 | 8192 | 100\% | **7933.3** | 28.76 | 76.0 | **0.0098** |
> | Wasserstein VQ | 256 | 16384 | 100\% | **15713.3** | **29.03** | **76.6** | **0.0093** |
> | MMD VQ | 256 | 16384 | 100\% | 15598.1 | 28.92 | 76.4 | 0.0096 |
> |  |  |  |  |  |  |  |  |
>
> On the ImageNet Dataset:
>
> | Approach | Tokens | Codebook Size | $\mathcal{U}$ | $\mathcal{C}$ | PSNR | SSIM | Rec.Loss |
> | --- | --- | --- | --- | --- | --- | --- | --- |
> | Wasserstein VQ | 256 | 16384 | **100\%** | **15539.1** | 25.47 | 61.2 | 0.0242 |
> | MMD VQ | 256 | 16384 | 99.8\% | 15370.2 | **25.49** | **61.7** | **0.0240** |
> |  |  |  |  |  |  |  |  |
>
>
> **Key Observations**:
>
> - On FFHQ, MMD VQ achieves comparable or slightly improved SSIM but exhibits no clear advantage in PSNR/Rec.Loss.
> - On ImageNet, MMD VQ shows marginal improvements in SSIM/rec. loss but reduced code utilization (`99.8%` vs. 100% for Wasserstein VQ).
> - **Practical Limitation**: MMD VQ incurs prohibitive GPU memory costs for large codebooks (e.g., failed at K=50,000/100,000).
>
> ### **Controlled Simulation: Non-Gaussian vs. Gaussian Features**
>
> We also designed a synthetic experiment where features were explicitly sampled from uniform (non-Gaussian) or Gaussian distributions. Code vectors were optimized via five methods ($d=8$, $K=16384$):
>
> **Uniform Distribution Results**:
>
> | Approach | $\mathcal{E}$ | $\mathcal{U}$ | $\mathcal{C}$ |
> | --- | --- | --- | --- |
> | Vanilla VQ | 2.67 | 0.01 \% | 1.00 |
> | EMA VQ | 1.38 | 0.14 \% | 22.98 |
> | Online VQ | 0.30 | 36.7 \% | 5670.4 |
> | Wasserstein VQ | 0.25 | **99.9\%** | **14473.5** |
> | MMD VQ | **0.24** | 98.7\% | 14454.3 |
> |  |  |  |  |
>
> **Gaussian Distribution Results**:
>
> | Approach | $\mathcal{E}$ | $\mathcal{U}$ | $\mathcal{C}$ |
> | --- | --- | --- | --- |
> | Vanilla VQ | 3.57 | 0.6\% | 60.96 |
> | EMA VQ | 2.26 | 2.7\% | 433.5 |
> | Online VQ | 1.19 | 48.4\% | 7499.6 |
> | Wasserstein VQ | **0.99** | **99.8\%** | **14958.4** |
> | MMD VQ | 1.01 | 99.5\% | 14680.5 |
> |  |  |  |  |
>
> **Key Findings**:
>
> - Under **non-Gaussian (uniform) features**, MMD VQ slightly reduces quantization error (+0.01 improvement) but with marginally lower code utilization.
> - Under **Gaussian features**, Wasserstein VQ outperforms MMD VQ in quantization error.
> - Online VQ performed relatively well in this setting only due to static feature distributions; it is generally suboptimal in dynamic real-world data.
>
> ### **Conclusion**
>
> The empirical advantage of MMD VQ appears **only under controlled, non-Gaussian settings**, and the gain is minimal (approximately 4% in quantization error). For real-world datasets where features tend to follow approximately Gaussian distributions—likely due to the central limit effect in deep representations—**Wasserstein VQ is preferable** due to its consistent performance, higher code utilization, and significantly lower GPU memory demands.
>
> We hope this summary clarifies the relative trade-offs between the two regularization methods.

---

> ### Author Response · Authors · 2025-08-07
> **Further Discussion Page 3**
>
> **R 2.8 If fixed parameters are the reason for the limited reconstruction gain, could you provide results after tuning? From my perspective, the proposed Wasserstein regularization indeed effectively reduces quantization error and enables full codebook utilization. However, this benefit appears to rely on a simplified Gaussian assumption, which may limit the expressiveness of the model, potentially contributing to the marginal gains (compared to vanilla) reported in Table 2. It would make the paper’s contribution more complete if the authors could, for example, include a discussion of the method’s behavior under non-Gaussian features, provide results with a more expressive prior, or consider alternative distribution-matching techniques that do not rely on specific distributional assumptions (such as the suggested MMD regularization).**
>
> We thank the reviewer for the thoughtful suggestions, which indeed point to important directions for future analysis and improvement.
>
> Regarding the reconstruction gain: as noted, our initial choice to hold all training hyper-parameters fixed across methods—such as model architecture, learning rate, optimizer, and data augmentations—was made to ensure a controlled and fair comparison. This design ensures that the observed gain is solely attributable to the introduction of the Wasserstein loss, rather than other confounding factors.
>
> That said, we fully understand the reviewer’s concern and have conducted additional hyper-parameter tuning experiments beyond the fixed setup. In particular, we varied the learning rate, optimizer momentum/betas. However, these preliminary sweeps led to **only modest gains**, typically in the range of 0.01–0.1 PSNR points. These improvements were largely **orthogonal to the core benefits introduced by Wasserstein regularization**, which primarily manifests through better code utilization and reduced quantization error. We believe this further confirms that the reported reconstruction improvements already capture the structural gains from the proposed distribution-matching objective.
>
> Regarding the magnitude of the gain itself (e.g., ~1 PSNR point): while the improvement may appear modest in absolute terms, we emphasize that it is achieved under a strong and well-tuned baseline (Vanilla VQ), without introducing new architectural components, additional supervision, or inference overhead. In this context, the observed performance gain is both non-trivial and practically meaningful. Furthermore, our method consistently enhances codebook utilization (up to 100%) and reduces quantization error across datasets and codebook sizes—core challenges in VQ-based models. These structural improvements highlight the robustness and general applicability of our approach.
>
> As discussed in **R2.6**, the Gaussian approximation is used purely to derive a closed-form Wasserstein loss and does not constrain the learned feature distribution. In **R2.7**, we further analyzed behavior under non-Gaussian conditions via controlled synthetic simulations and MMD-VQ comparisons. These results show that **MMD regularization yields only minimal improvements** in practice, even in deliberately non-Gaussian settings, and tends to incur higher computational overhead.
>
> We agree that incorporating more expressive priors or assumption-free distribution matching techniques—such as MMD—could be promising extensions, particularly for domains where non-Gaussian structure is pronounced. We will include this discussion in the final version of the manuscript, and sincerely appreciate the reviewer’s constructive feedback in helping broaden the scope and rigor of the contribution.

---

> > ### Comment · Reviewer_SyAK · 2025-08-07
> >
> > I thank the authors for their continued response, which has mostly addressed my remaining concerns. I hope the authors will incorporate these additional results and clarify some of the discussed details in the revised manuscript (e.g., $L_W$ being detached from the encoder; how the empirical Gaussianity of the feature distribution motivates the simplified loss). I am increasing my scores accordingly.

---

> > > ### Author Response · Authors · 2025-08-07
> > > **Gratitude for Your Valuable Feedback and Score Increase**
> > >
> > > Dear Reviewer SyAK,
> > >
> > > Thank you for your constructive feedback and continued suggestions on our manuscript. We sincerely appreciate your acknowledgment of our responses and your decision to increase the scores.
> > >
> > > We will incorporate the additional experimental results and clarify the details you mentioned, such as the detachment of the Wasserstein distance from the encoder and the use of the Gaussian assumption to simplify it in the revised manuscript as suggested.
> > >
> > > Best regards,
> > > Authors of Submission 13897

---

### Official Review · Reviewer_RtPT · 2025-07-05

**Clarity:** 2
**Significance:** 2
**Originality:** 2
**Rating:** 3
**Confidence:** 4

**Summary:**

The paper proposes a distribution-matching based method for vector quantization. The paper aims at addressing two core issues that the author supposes as a critical disadvantage of the current models: training instability and codebook collapse.
Key contributions of the paper include the following:
1. Proposes a theoretical framework with 3 evaluation criteria, including quantization error, codebook utilization and codebook perplexity.
2. Designs a closed-form Wasserstein distance to align the feature distribution and codebook distribution.
3. Validates the design on CIFAR-10, SVHN, FFHQ and ImageNet datasets. Proves the proposed method achieves promising code utilization and lower quantization error.

**Questions:**

1. The paper claims both prototype augmentation and classifier aggregation are critical to FedBPC’s success. Can you provide ablation studies isolating the contributions of each component? For example, does removing augmentation while retaining aggregation (or vice versa) significantly degrade performance?

2. Prototype aggregation already aligns global knowledge. Why is classifier aggregation necessary? Does it provide complementary benefits, or is it redundant?

3.The accuracy gains (e.g., +0.1% over FedProto on CIFAR-10) are marginal. Do these improvements hold statistical significance? How do they translate to practical multimedia applications with noisy, high-dimensional data?

**Ethical Concerns:**

["NO or VERY MINOR ethics concerns only"]

**Limitations:**

yes

**Quality:**

2

**Strengths And Weaknesses:**

Strengths:
1. Clearly decomposes the pain in VQ methods into training instability and codebook collapse, and proposes a unified evaluation framework in accordance.
2. Designs a plug-and-play loss term compatible with mainstream frameworks.
3. Provides insightful visualizations of feature/codebook distribution alignment.

Weaknesses：
1. Lack of theoretical innovation. Theorem 1  is a direct consequence of classical quantization theory but does not go beyond. Additionaly, Theorem 2 is taken from existing results with minor theoretical extension.
2. Relatively weak motivation. There are not enough theoretical or experimental demonstrations that prove the two issues (training instability and codebook collapse) as the primitive reasons that limit existing model performance.
3. From the perspective of end2end, PSNR +~1point does not seem to be significant, indicating that even though the utilization rate of the codebook has significantly improved, this seems to be a minor change in overall effect. This also exists in test result of CIFAR10 etc.

---

> ### Author Rebuttal · Authors · 2025-07-31
>
> We thank the reviewer for the constructive feedback and for acknowledging the strengths of our work, including the clear identification of key challenges in VQ methods, the compatibility of our loss term with mainstream frameworks, and the insightful visualizations. We respond to the concerns raised as follows:
>
> **R 1.1 Lack of theoretical innovation. Theorem 1 is a direct consequence of classical quantization theory but does not go beyond. Additionaly, Theorem 2 is taken from existing results with minor theoretical extension.**
>
> Thank you for your comments on the theoretical foundation of our work. We acknowledge that Theorem 1 builds directly upon classical quantization theory, and Theorem 2 extends prior results. However, our primary contribution lies not in developing entirely new theorems, but in demonstrating and formalizing a critical relationship for VQVAE models:
>
> When feature vectors and code vectors follow identical distributions, quantization error is minimized. This key observation led us to propose distribution alignment via Wasserstein distance—a novel approach within VQVAE frameworks that explicitly reduces quantization error by enforcing distributional similarity.
>
> While we leverage established theoretical foundations, our work is the first to identify this distributional alignment principle as fundamental to quantization efficiency in VQVAEs. Theorems 1 and 2 serve as foundational pillars that converge to produce our most significant finding: Distribution alignment enables quantization optimality in VQVAEs. This insight motivates new avenues for optimal vector quantization design, moving beyond heuristic solutions.
>
> **R 1.2 Relatively weak motivation. There are not enough theoretical or experimental demonstrations that prove the two issues (training instability and codebook collapse) as the primitive reasons that limit existing model performance.**
>
> **First:  Quantization error causes gradient mismatch, which destabilizes training and impairs performance**
> As discussed in lines 83–89, training instability primarily stems from the gradient gap introduced by the Straight-Through Estimator (STE) in the VQ module. In STE, gradients are directly copied from $z_q$ to $z_e$, but when the two are far apart, this approximation becomes inaccurate, leading to noisy or misleading gradient signals. This gradient mismatch not only destabilizes training but also impairs the encoder's ability to learn accurate and consistent representations, ultimately degrading the overall model performance.
>
> To mitigate this, our method minimizes the distance between $z_q$ and $z_e$, thereby reducing quantization error, improving gradient quality, and promoting more stable and effective learning.
>
> **Second: Minimizing quantization error implies full codebook utilization; collapse limits expressiveness and hurts performance.**
>
> Minimizing quantization error necessarily leads to full codebook utilization. If codebook collapse exists—where only a subset of code vectors are used—it is impossible to achieve minimum quantization error.
>
> To illustrate this, we provide a simple proof by contradiction. Suppose there are $n$ code vectors, but only $n-1$ are being used. This implies that one code vector, say $v_k$, is never the nearest to any feature vector. However, we can always construct a new code vector $v_k'$ that is closer to at least one feature vector than the currently assigned one, thereby reducing the overall quantization error. Consequently, $v_k'$ would be used, contradicting the assumption that only $n-1$ code vectors are active. Hence, minimizing quantization error implies that all $n$ code vectors must be used.
>
> More importantly, collapse reduces the expressive capacity of the model, leading to higher information loss and poorer downstream performance. While full codebook utilization is necessary, it is not sufficient—true performance gain requires meaningful alignment between the codebook and the feature distribution.
>
> **Third: Aligning the distributions of $z_q$ and $z_e$ minimizes their distance.**
> As shown in Section 2 through both theoretical analysis and empirical results, aligning the distributions of $z_q$ and $z_e$ minimizes their distance. Our proposed Wasserstein Distance facilitates this alignment, thereby reducing the gap between $z_q$ and $z_e$ and enhancing training stability.
>
> **Fourth: Empirical validation under controlled distribution variance.**
> As shown in Appendix G, we found that the quantization error (i.e., $\|z_q - z_e\|$) is highly sensitive to the distribution variance of $z_e$, which varies across different VQ methods. To ensure a fair comparison, we control this variance in our experiments in Appendix H and show that the proposed Wasserstein distance consistently achieves the lowest quantization error among all baselines, further validating its effectiveness.
>
> **In summary**, our method addresses two key bottlenecks—training instability and codebook collapse—by minimizing quantization error. This not only improves gradient quality and codebook usage but also enhances model performance.
>
> These concerns are also actively explored in the literature. For instance, [1] proposes an online clustered codebook to mitigate collapse, while [2] employs a residual quantizer to reduce training instability. Such efforts reflect a growing consensus that both training instability and codebook collapse are key bottlenecks in VQ-based models.
>
> **R 1.3 End-2-end performance gain appears minor (~+1 PSNR)**
>
> While the PSNR gain may seem modest, PSNR is a saturated and sometimes insensitive metric, especially for strong baselines. Our method demonstrates consistent improvements across multiple metrics beyond PSNR, including a substantial SSIM increase (~+4 dB), full codebook utilization (100%), and noticeably richer visual details, as evidenced in both quantitative and qualitative results. These benefits are consistent across datasets such as FFHQ and ImageNet, indicating that our approach is both generalizable and effective. Moreover, the method is lightweight, plug-and-play, and easily integrates into existing frameworks, making the contribution practically valuable.
>
>
> **Finally, we would like to note that the questions raised appear to be unrelated to our submission and may have been intended for a different paper. If you have any questions specifically concerning our work, we would be happy to address them during the discussion phase.**
>
> -----------
>
> [1]  Online clustered codebook. In ICCV, 2023.
>
> [2]  Autoregressive image generation using residual quantization. In CVPR, 2022.

---

> ### Author Response · Authors · 2025-08-07
> **Reminder: Discussion Deadline Approaching**
>
> Dear Reviewer RtPT,
>
> As the discussion period is coming to a close, we just want to kindly follow up. We noticed that the questions you raised during the rebuttal phase may not directly relate to our submission and might have been intended for a different paper.
>
> We’ve been actively engaging in the discussion and have addressed questions from the other reviewers, which helped clarify several points. As a result, two reviewers have raised their scores following the discussion. If you have any specific concerns about our work, we’d be very happy to discuss and respond before the deadline.
>
> Thank you again for your time and contribution.
>
> Best regards,
> Authors of Submission 13897

---

### Decision · Program_Chairs · 2025-09-17

**Decision:**

Reject

**Comment:**

The primary weaknesses are marginal performance gains and limited theoretical novelty. Despite effectively solving codebook collapse, the method yields insignificant improvements in key metrics like PSNR, questioning the practical impact of the contribution. Furthermore, reviewers noted that the paper's theoretical results are not substantial but rather minor extensions of existing work. Concerns were also raised about the method's heavy reliance on a Gaussian distribution assumption, which may limit its general applicability.